# GRASSMANNIAN CLASS REPRESENTATION IN DEEP LEARNING

## ABSTRACT

We generalize the class representative vector found in deep classification networks to linear subspaces and show that the new formulation enables the simultaneous enhancement of the inter-class discrimination and intra-class feature variation. Traditionally, the logit is computed by the inner product between a feature and the class vector. In our modeling, classes are subspaces and the logit is defined as the norm of the projection from a feature onto the subspace. Since the set of subspaces forms Grassmann manifolds, finding the optimal subspace representation for classes is to optimize the loss on a Grassmannian. We integrate the Riemannian SGD into existing deep learning frameworks such that the class subspaces in a Grassmannian are jointly optimized with other model parameters in Euclidean. Compared to the vector form, subspaces have two appealing properties: they can be multi-dimensional and they are scaleless. Empirically, we reveal that these distinct characteristics improve various tasks. (1) Image classification. The new formulation brings the top-1 accuracy of ResNet50-D on ImageNet-1K from 78.04% to 79.37% using the standard augmentation in 100 training epochs. This confirms that the representative capability of subspaces is more powerful than vectors. (2) Feature transfer. Subspaces provide freedom for features to vary and we observed that the intra-class variability of features increases when the subspace dimensions are larger. Consequently, the quality of features is better for downstream tasks. The average transfer accuracy across 6 datasets improves from 77.98% to 80.12% compared to the strong baseline of vanilla softmax. (3) Long-tail classification. The scaleless property of subspaces benefits classification in the long-tail scenario and improves the accuracy of ImageNet-LT from 46.83% to 48.94% compared to the standard formulation. With these encouraging results, we believe that more applications could benefit from the Grassmannian class representation. Codes will be released.

## 1 INTRODUCTION

The idea of representing classes as linear subspaces in machine learning can be dated back, at least, to 1973 (Watanabe & Pakvasa (1973)), yet it is mostly ignored in the current deep learning literature. In this paper, we revisit the scheme of representing classes as linear subspaces in the deep learning context. To be specific, each class $i$ is associated with a linear subspace $S_i$, and for any feature vector $\boldsymbol{x}$, the $i$-th class logit is defined as the norm of projection

$$l_i := \left\| \operatorname{proj}_{S_i} \boldsymbol{x} \right\|. \tag{1}$$

Since a subspace is a point in the Grassmann manifold (Absil et al. (2009)), we call this formulation the Grassmannian class representation. In the following, we answer the two critical questions,

1. Is Grassmannian class representation useful in real applications?

2. How to optimize the subspaces in training?

The procedure `fully-connected layer` $\to$ `softmax` $\to$ `cross-entropy loss` is the standard practice in deep classification networks. Each column of the weight matrix of the fully-connected layer is called the class representative vector and serves as a prototype for one class. This representation of class has achieved huge success, yet it is not without imperfections.

In the study of transferable features, researchers noticed a dilemma that representations with higher classification accuracy on the original task lead to less transferable features for downstream tasks (Kornblith et al. (2021); Müller et al. (2019)). This is connected to the fact that they tend to collapse intra-class variability of representations, resulting in loss of information in the logits about the resemblances between instances of different classes. Furthermore, the neural collapse phenomenon (Papyan et al. (2020)) indicates that as training progresses, the intra-class variation becomes negligible, and features collapse to their class-means. So this dilemma inherently originates from the practice of representing classes by a single vector. The Grassmannian class representation shed light on this issue as features of each class are allowed to vary in a high-dimensional subspace without incurring losses in classification.

In the study of the long-tail classification, researchers found that the norm of class representative vectors is highly related to the number of training instances in the corresponding class (Kang et al. (2019)) and the recognition accuracy is affected. To counter this effect, the class representative vector is typically been rescaled to unit length during training (Liu et al. (2019)) or re-calibrated in an extra post-processing step (Kang et al. (2019)). In addition to these techniques, the Grassmannian class representation provides a natural and elegant solution for this as subspace is scaleless.

It is well known that the set of $k$-dimensional linear subspaces form a Grassmann manifold, so finding the optimal subspace representation for classes is to optimize on the Grassmann manifold. Thus for the second question, the natural solution is to use the geometric optimization (Edelman et al. (1998)), which optimizes an objective function under the constraint of a given manifold. Points being optimized are moving along geodesics instead of following the direction of Euclidean gradients. The preliminary concepts of geometric optimization are reviewed in Section 3, and the technical details of subspace learning are presented in Section 4. We implemented an efficient Riemannian SGD for optimization in Grassmann manifold as shown in Algorithm 1, which integrates the geometric optimization algorithms to deep learning frameworks so that both the linear subspaces in Grassmannian and model weights in Euclidean are jointly optimized.

Going back to the first question, we experiment on three concrete tasks in Section 5 to demonstrate the practicality and effectiveness of Grassmannian class representation. We find that (1) Grassmannian class representation improves large-scale image classification accuracy. (2) Grassmannian class representation produces high-quality features that can better transfer to downstream tasks. (3) Grassmannian class representation improves the long-tail classification accuracy. With these encouraging results, we believe that Grassmannian class representation is a promising formulation and more applications may benefit from its attractive features.

## 2 RELATED WORK

**Geometric Optimization**   Edelman et al. (1998) developed the geometric Newton and conjugate gradient algorithms on the Grassmann and Stiefel manifolds in their seminal paper. Riemannian SGD was introduced in Bonnabel (2013) with an analysis on convergence and there are variants such as Riemannian SGD with momentum (Roy et al. (2018)) or adaptive (Kasai et al. (2019)). Other popular Euclidean optimization methods such as Adam are also studied in the Riemannian manifold context (Becigneul & Ganea (2019)). Lezcano-Casado & Martınez-Rubio (2019) study the special case of $SO(n)$ and $U(n)$ and uses the exponential map to enable Euclidean optimization methods for Lie groups. The idea was generalized into trivialization in Lezcano Casado (2019). Our Riemannian SGD Algorithm 1 is tailored for Grassmannian, so we have a closed-form equation for geodesics. Other applications of geometric optimization include matrix completion (Mishra & Sepulchre (2014); Li et al. (2015b;a); Nimishakavi et al. (2018)), hyperbolic taxonomy embedding (Nickel & Kiela (2018)), etc. Hamm & Lee (2008) propose the Grassmann discriminant analysis, in which features are modeled as linear subspaces. These applications are mostly using *shallow* models. Zhang et al. (2018) use subspaces to model clusters in unsupervised learning, which share similar spirit with our work. Simon et al. (2020) model classes as subspaces in few-shot learning, however, their subspaces are computed from data matrix rather than explicitly parametrized and learned. Roy et al. (2019) use Stiefel manifold to construct Mahalanobis distance matrix in Siamese networks in order to improve feature embeddings of deep metric learning.

**Orthogonal Constraints in Deep Learning**  There are works that enforce orthogonality on weights, which study the regularization effect of orthogonal constraints. Contrastingly, we used orthogonal matrices as the numerical representation of the geometry object of subspaces and focus on the representation of classes. The approaches of enforcing orthogonality include regularizations (Arjovsky et al. (2016); Xie et al. (2017a); Bansal et al. (2018); Qi et al. (2020); Wang et al. (2020), etc.), geometric constraints (Ozay & Okatani (2018); Harandi & Fernando (2016)) and paraunitary systems (Su et al. (2022)). Orthogonally constrained data is also explored by Huang et al. (2018).

**Improving Diversity in Feature Learning**  Grassmannian class representation encourages the intra-class variation *implicitly* by providing a subspace to vary. In metric learning, there are efforts to *explicitly* encourage feature diversity. For example, SoftTriplet Loss (Qian et al. (2019)) models each class as local clusters with several centers. Zhang et al. (2017) use a global orthogonal regularization to encourage local descriptors to spread out in the features space. Yu et al. (2020) propose to learn low-dimensional structures from the maximal coding rate reduction principle. The subspaces are estimated using PCA on feature vectors after the training. In our formulation, subspaces are directly optimized in the Grassmann manifold during training.

**Normalized Classification Weights**  Normalizing class representative vectors has been found useful in representation learning (Wang et al. (2017; 2018); Deng et al. (2019)) and long-tail classification (Liu et al. (2019); Wang et al. (2021)). However, works such as ArcFace (Deng et al. (2019)) focus on adding an extra margin to suppress intra-class variance. In contrast, our subspace formulation encourages intra-class variation.

## 3 PRELIMINARIES

In this section, we briefly review the essential concepts in geometric optimization. Detailed exposition can be found in Edelman et al. (1998) and Absil et al. (2009). Given an $n$-dimensional Euclidean space $\mathbb{R}^n$, the set of $k$-dimensional linear subspaces forms the Grassmann manifold $\mathcal{G}(k, n)$. A computational-friendly representation for subspace $S \in \mathcal{G}(k, n)$ is an orthonormal matrix $\boldsymbol{S} \in \mathbb{R}^{n \times k}$, where $\boldsymbol{S}^T \boldsymbol{S} = \boldsymbol{I}_k$ and $\boldsymbol{I}_k$ is the $k \times k$ identity matrix. Columns of matrix $\boldsymbol{S}$ can be interpreted as an orthonormal basis for the subspace $S$. The matrix representation is *not unique*, as right multiplying by an orthonormal matrix will get a new matrix representing the same subspace. Formally, Grassmannian is a quotient space of the Stiefel manifold and the orthogonal group $\mathcal{G}(k, n) = \mathrm{St}(k, n)/\mathcal{O}(k)$, where $\mathrm{St}(k, n) = \{\boldsymbol{X} \in \mathbb{R}^{n \times k} | \boldsymbol{X}^T \boldsymbol{X} = \boldsymbol{I}_k\}$ and $\mathcal{O}(k) = \{\boldsymbol{X} \in \mathbb{R}^{k \times k} | \boldsymbol{X}^T \boldsymbol{X} = \boldsymbol{I}_k\}$. When the context is clear, we use the notation of space $S$ and one of its matrix representations $\boldsymbol{S}$ interchangeably. The tangent space of the Grassmann manifold at $S$ consists of all $n \times k$ matrices $\boldsymbol{T}$ such that $\boldsymbol{S}^T \boldsymbol{T} = \boldsymbol{0}$.

Given a function $f : \mathcal{G}(k, n) \to \mathbb{R}$ defined on the Grassmann manifold, the Riemannian gradient of $f$ at point $S \in \mathcal{G}(k, n)$ is given by (Edelman et al., 1998, Equ. (2.70)),

$$\nabla f(\boldsymbol{S}) = f_{\boldsymbol{S}} - \boldsymbol{S}\boldsymbol{S}^T f_{\boldsymbol{S}}, \tag{2}$$

where $f_{\boldsymbol{S}}$ is the Euclidean gradient with elements $(f_{\boldsymbol{S}})_{ij} = \frac{\partial f}{\partial \boldsymbol{S}_{ij}}$. When performing gradient descend on the Grassmann manifold, and suppose the current point is $\boldsymbol{S}$ and the current Riemannian gradient is $\boldsymbol{G}$, then the next point is the endpoint of $\boldsymbol{S}$ moving along the geodesic toward the tangent $\boldsymbol{G}$ with some step size. The formula of the geodesic is given by (Edelman et al., 1998, Equ. (2.65)),

$$\boldsymbol{S}(t) = (\boldsymbol{S}\boldsymbol{V}\cos(t\boldsymbol{\Sigma}) + \boldsymbol{U}\sin(t\boldsymbol{\Sigma}))\,\boldsymbol{V}^T, \tag{3}$$

where $\boldsymbol{U}\boldsymbol{\Sigma}\boldsymbol{V}^T = \boldsymbol{G}$ is the thin singular value decomposition of $\boldsymbol{G}$.

## 4 LEARNING THE GRASSMANNIAN CLASS REPRESENTATION

Denote the weight of the last fully-connected layer in a classification network by $\boldsymbol{W} \in \mathbb{R}^{n \times C}$ and the bias by $\boldsymbol{b} \in \mathbb{R}^C$, where $n$ is the dimension of features and $C$ is the number of classes. The $i$-th column vector $\boldsymbol{w}_i$ of $\boldsymbol{W}$ is called the $i$-th class representative vector. The $i$-th logit is computed as the inner product between a feature $\boldsymbol{x}$ and the class vector (and optionally offset by a bias $b_i$), namely $\boldsymbol{w}_i^T \boldsymbol{x} + b_i$. We extend this well-established formula to a multi-dimensional subspace form

$$l_i := \left\|\mathrm{proj}_{S_i}\boldsymbol{x}\right\|, \tag{4}$$

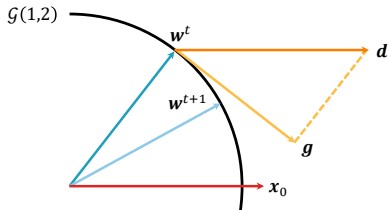

Figure 1: Geometric optimization in Grassmann manifold $\mathcal{G}(1,2)$. Each point (*e.g.* $\boldsymbol{w}^t$) in the black circle represent the 1-dimensional linear subspace passing through it. $\boldsymbol{g}$ is the Riemannian gradient obtained by the projection of Euclidean gradient $\boldsymbol{d}$. $\boldsymbol{w}^t$ moves along the geodesic towards the direction $\boldsymbol{g}$ to a new point $\boldsymbol{w}^{t+1}$.

where $S_i \in \mathcal{G}(k, n)$ is a $k$-dimensional subspace in the $n$-dimensional feature space. We call $S_i$ the $i$-th *class representative space*, or class space in short. Comparing the new logit to the standard one, the inner product of feature $\boldsymbol{x}$ with class vector is replaced by the norm of the subspace projection $\mathrm{proj}_{S_i} \boldsymbol{x}$ and the bias term is omitted. We found that re-normalizing features to a constant length $\gamma$ improves training. Incorporating this, Equation (4) becomes $\left\| \mathrm{proj}_{S_i} \frac{\gamma \boldsymbol{x}}{\|\boldsymbol{x}\|} \right\|$. To simplify notation, we assume feature $\boldsymbol{x}$ has been properly re-normalized throughout this paper unless otherwise specified.

The application of the subspace class representation requires two modifications to an existing network. Firstly, the last fully-connected layer is replaced by its geometric counterpart, which is detailed in Section 4.1. The new geometric layer will transform features to logits using Equation (4). Secondly, the optimizer should be extended to process the new geometric layer simultaneously, which is explained in Section 4.2. Parameters of the geometric layer are optimized using Geometric SGD, while all other parameters are optimized as usual using the standard SGD algorithm.

## 4.1 GRASSMANNIAN CLASS REPRESENTATION

Suppose for class $i$, $i = 1, 2, \ldots, C$, its subspace representation is $S_i \in \mathcal{G}(k_i, n)$, where the dimension $k_i$ is a hyperparameter and is fixed during training. Then the tuple of subspaces $(S_1, S_2, \ldots, S_C)$ will be optimized in the product space $\mathcal{G}(k_1, n) \times \mathcal{G}(k_2, n) \times \cdots \times \mathcal{G}(k_C, n)$. Denote a matrix instantiation of $S_i$ as $\boldsymbol{S}_i \in \mathbb{R}^{n \times k}$, where the column vectors form an orthonormal basis $S_i$, then we concatenate the matrices into a big matrix

$$\boldsymbol{S} = [\boldsymbol{S}_1 \; \boldsymbol{S}_2 \; \cdots \; \boldsymbol{S}_C] \in \mathbb{R}^{n \times (k_1 + k_2 + \cdots + k_C)}. \tag{5}$$

The matrix $\boldsymbol{S}$ contains the parameters that are optimized numerically. For feature $\boldsymbol{x}$, the product $\boldsymbol{S}_i^T \boldsymbol{x}$ gives the coordinate of $\mathrm{proj}_{S_i} \boldsymbol{x}$ under the orthonormal basis formed by the columns of $\boldsymbol{S}_i$. By definition in Equation (4), the logit for class $i$ and feature $\boldsymbol{x}$ is computed by

$$l_i = \left\| \mathrm{proj}_{S_i} \boldsymbol{x} \right\| = \left\| \boldsymbol{S}_i^T \boldsymbol{x} \right\|. \tag{6}$$

**Grassmannian Fully-Connected Layer**  We can implement a geometric fully-connected layer using the plain old fully-connected layer. The shape of the weight $\boldsymbol{S}$ is $n \times (k_1 + k_2 + \cdots + k_C)$, as shown in Equation (5). In the forward pass, the input feature is multiplied with the weight matrix to get a temporary vector $\boldsymbol{t} = \boldsymbol{S}^T \boldsymbol{x}$, then the first element of the output is the norm of the sub-vector $(t_1, \ldots, t_{k_1})$, and the second element of the output is the norm of $(t_{k_1+1}, t_{k_1+2}, \ldots, t_{k_1+k2})$, etc.

**Parameter Initialization**  Each matrix instantiation of the subspace should be initialized as an orthonormal matrix. The geometric optimization algorithm described in Section 4.2 ensures their orthonormality during training. Specifically, for Grassmannian fully-connected layer, each block $\boldsymbol{S}_i$ of the weight $\boldsymbol{S}$ in Equation (5) is orthonormal. The whole matrix $\boldsymbol{S}$ needs not be orthonormal.

## 4.2 OPTIMIZE THE SUBSPACES

Geometric optimization is to optimize functions defined on manifolds. The key step is to find the Riemannian gradient of the loss function and then descend along the geodesic. Here the manifold in concern is the Grassmannian $\mathcal{G}(k, n)$. As an intuitive example, $\mathcal{G}(1, 2)$ consists of all lines through the origin in a two-dimensional plane. We can visualize it as a unit circle where each point on the unit circle represents the line passing through it. Antipodal points represent the same line. To illustrate

---

**Algorithm 1** An Iteration of the Riemannian SGD with Momentum for Grassmannian at Step $t$

---

**Input:** Learning rate $\gamma > 0$, momentum $\mu \in [0, 1)$, Grassmannian weight matrix $\boldsymbol{S}^{(t)} \in \mathbb{R}^{n \times k}$, momentum buffer $\boldsymbol{M}^{(t-1)} \in \mathbb{R}^{n \times k}$, Euclidean gradient $\boldsymbol{D} \in \mathbb{R}^{n \times k}$.

1: Compute Riemannian gradient $\boldsymbol{G} \leftarrow (\boldsymbol{I}_n - \boldsymbol{S}\boldsymbol{S}^T)\boldsymbol{D}$.          ▷ Equation (8)
2: Approximately parallel transport $\boldsymbol{M}$ to the tangent space of current point $\boldsymbol{S}^{(t)}$ by projection
$$\boldsymbol{M} \leftarrow (\boldsymbol{I}_n - \boldsymbol{S}\boldsymbol{S}^T)\boldsymbol{M}^{(t-1)}. \tag{11}$$
3: New momentum $\boldsymbol{M}^{(t)} \leftarrow \mu\boldsymbol{M} + \boldsymbol{G}$.          ▷ PyTorch version
4: Move along geodesic using equation (3). If $\boldsymbol{U}\boldsymbol{\Sigma}\boldsymbol{V}^T = \boldsymbol{M}^{(t)}$ is the thin singular value decomposition, then
$$\boldsymbol{S}^{(t+1)} \leftarrow \left(\boldsymbol{S}^{(t)}\boldsymbol{V}\cos(\gamma\boldsymbol{\Sigma}) + \boldsymbol{U}\sin(\gamma\boldsymbol{\Sigma})\right)\boldsymbol{V}^T.$$
5: (Optional) Re-orthogonalization $\boldsymbol{S}^{(t+1)}$ by QR decomposition.          ▷ For numerical stability

---

how geometric optimization works, we define a toy problem on $\mathcal{G}(1, 2)$ that maximizes the norm of the projection of a fixed vector $\boldsymbol{x}_0$ onto a line through the origin, namely

$$\max_{S \in \mathcal{G}(1,2)} \|\mathrm{proj}_S \boldsymbol{x}_0\|. \tag{7}$$

As shown in Figure 1, we represent $S$ with a unit vector $\boldsymbol{w} \in S$. Suppose at step $t$, the current point is $\boldsymbol{w}^{(t)}$, then it is easy to compute that the Euclidean gradient at $\boldsymbol{w}^{(t)}$ is $\boldsymbol{d} = \boldsymbol{x}_0$, and the Riemannian gradient $\boldsymbol{g}$ is the Euclidean gradient $\boldsymbol{d}$ projected to the tangent space of $\mathcal{G}(1, 2)$ at point $\boldsymbol{w}^{(t)}$. The next iterative point $\boldsymbol{w}^{(t+1)}$ is to move $\boldsymbol{w}^{(t)}$ along the geodesic toward the direction $\boldsymbol{g}$. Without geometric optimization, the next iterative point would have lied at $\boldsymbol{w}^{(t)} + \gamma\boldsymbol{d}$, jumping outside of the manifold.

The following proposition computes the Riemannian gradient we needed.

**Proposition 1.** *Let $\boldsymbol{S} \in \mathbb{R}^{n \times k}$ be a matrix instantiation of subspace $S \in \mathcal{G}(k, n)$, and $\boldsymbol{x} \in \mathbb{R}^n$ is a vector in Euclidean space, then the Riemannian gradient $\boldsymbol{G}$ of $l(S, \boldsymbol{x}) = \|\mathrm{proj}_S \boldsymbol{x}\|$ w.r.t. $S$ is*

$$\boldsymbol{G} = \frac{1}{l}(\boldsymbol{I}_n - \boldsymbol{S}\boldsymbol{S}^T)\boldsymbol{x}\boldsymbol{x}^T\boldsymbol{S}. \tag{8}$$

*Proof.* Rewrite $\|\mathrm{proj}_S \boldsymbol{x}\| = \sqrt{\boldsymbol{x}^T \boldsymbol{S}\boldsymbol{S}^T \boldsymbol{x}}$, and compute the Euclidean derivatives as

$$\frac{\partial l}{\partial \boldsymbol{S}} = \frac{1}{l}\boldsymbol{x}\boldsymbol{x}^T\boldsymbol{S}, \quad \frac{\partial l}{\partial \boldsymbol{x}} = \frac{1}{l}\boldsymbol{S}\boldsymbol{S}^T\boldsymbol{x}. \tag{9}$$

Then Equation (8) follows from Equation (2).          $\square$

We give a geometric interpretation of Proposition 1. Let $\boldsymbol{w}_1$ be the unit vector along direction $\mathrm{proj}_S \boldsymbol{x}$, then expand it to an orthonormal basis of $S$, say $\{\boldsymbol{w}_1, \boldsymbol{w}_2, \ldots, \boldsymbol{w}_k\}$. Since Riemannian gradient is invariant to the matrix instantiation, we can set $\boldsymbol{S} = [\boldsymbol{w}_1 \ \boldsymbol{w}_2 \ \cdots \ \boldsymbol{w}_k]$. Then Equation (8) becomes

$$\boldsymbol{G} = \begin{bmatrix} (\boldsymbol{I}_n - \boldsymbol{S}\boldsymbol{S}^T)\boldsymbol{x} & \boldsymbol{0} & \cdots & \boldsymbol{0} \end{bmatrix}, \tag{10}$$

since $\boldsymbol{w}_i \perp \boldsymbol{x}, i = 2, 3, \ldots, k$ and $\boldsymbol{w}_1^T\boldsymbol{x} = l$. Equation (10) shows that in the single-sample case, only one basis vector $\boldsymbol{w}_1$ needs to be rotated towards vector $\boldsymbol{x}$, where $\boldsymbol{w}_1$ is the unit vector in $S$ that is closest to $\boldsymbol{x}$.

**Riemannian SGD**  During training, parameters of non-geometric layers are optimized as usual using the vanilla SGD algorithm. For geometric layers such as the Grassmannian fully-connected layer, their parameters are optimized using the Riemannian SGD algorithm. The pseudo-code of the Riemannian SGD with momentum, which we implemented in our experiments, is described in Algorithm 1. We only show the code for the single-sample, single Grassmannian case. It is trivial to extend them to the batch version and the product of Grassmannians. Note that in step 2, we use projection to approximate the parallel translation of momentum for efficiency, and in step 5 an optional extra orthogonalization can improve numerical stability. The momentum update formula is adapted from the PyTorch implementation of the vanilla SGD. Weight decay does not apply here since spaces are scaleless. Algorithm 1 works together with the vanilla SGD and modifies the gradient from Euclidean to Grassmannian on-the-fly for geometric parameters.

## 5 EXPERIMENT

In this section, we study the influence of Grassmannian class representation through experiments. Firstly, in Section 5.1, we show that the expressive power of Grassmannian class representation improves accuracy in large-scale image classification. Secondly, in Section 5.2, we show that the Grassmannian class representation improves the feature transferability by allowing larger intra-class variation. Thirdly, in Section 5.3, we demonstrated that the scaleless property of the Grassmannian class representation improves the classification accuracy in the long-tail scenario. Additional experiments on hyper-parameter choices and design decisions are presented in Appendix B.

We choose the vanilla softmax loss and the cosine softmax loss (without margin) as baselines since they reflect the current typical class representations. The former uses a plain vector and the latter uses a normalized vector. Other innovations on losses, such as adding margins (Deng et al. (2019)), re-balancing class-wise gradients (Wang et al. (2021)), are orthogonal to our contribution.

### 5.1 GRASSMANNIAN CLASS REPRESENTATION IMPROVES CLASSIFICATION ACCURACY

We apply the Grassmannian class representation to large-scale classification, where consistent improvement over baselines is shown. We then analyze the characteristics of both the learned features and the learned class subspaces. On the feature representation side, we compare the feature sparsity and intra-class variability. On the class representation side, we visualize the principal angles between any pair of classes, a concept that only appears when classes are Grassmannian.

**Experimental Setting** We use the ResNet50-D (He et al. (2019)) architecture as the base model, and benchmark on ImageNet-1K (Deng et al. (2009)). ResNet50-D is a slight modification of the original ResNet-50 (He et al. (2016)) with about 1% improvement in accuracy. ImageNet-1K is a large-scale image classification dataset containing 1.28M training images and 50K validation images in 1000 categories. We set $\gamma = 25$ for both cosine softmax and the Grassmannian class representation. Our method replaces the last fully-connected layer of ResNet50-D by a Grassmannian fully-connected layer. To reduce the number of hyper-parameters, we simply set the subspace dimension $k$ to be the same for all classes. We vary the hyper-parameter $k$ in the range $[1, 2, 4, 8, 16]$. Since the dimension of feature is 2048, the Grassmannian fully-connected layer has the geometry of $\Pi_{i=1}^{1000} \mathcal{G}(k, 2048)$.

**Training Strategy** All settings share the same training strategy. Each training includes 100 epochs with total batch size 256 on 8 NVIDIA Tesla V100 GPUs. SGD is used for baselines and Riemannian SGD described in Algorithm 1 is used for Grassmannian class representations. The momentum is 0.9 and the weight decay is 0.0001. The initial learning rate is 0.1 and then follows the cosine learning rate decay. The checkpoint with best validation score is used. The input size is $224 \times 224$ and we use the standard augmentation for ImageNet, namely, random resized crop followed by random horizontal flip. The code is implemented using the *mmclassification* (MMClassification Contributors (2020)) package, and uses PyTorch as the training backend. Note that to make the number of experiments tractable due to our limited computation resources, we omitted many tricks that has shown to improve representation learning, such as stronger augmentation (Cubuk et al. (2020)), longer training (Wightman et al. (2021)), adding margins (Deng et al. (2019)) etc., and focus on the improvements solely contributed by the Grassmannian formulation.

**Feature Norm Regularization** We noticed that the norm of the feature (before re-normalization) decreases as training progresses (details see Appendix A). For example, in the case of $k = 16$, the average norm of feature decreases from 1.051 at epoch 10 to 0.332 at epoch 100. Although the norm of the feature does not affect inference result due to the feature re-normalization when computing logits, we empirically find that encouraging the norm to be larger than a constant $L$ improves the training. Specifically, we propose a feature norm regularization loss $L^{\text{FN}}$,

$$L^{\text{FN}} = \frac{1}{K} \sum_i \frac{1}{2} \left( \text{relu} \left( L - \|\boldsymbol{x}_i\| \right) \right)^2, \tag{12}$$

where $\boldsymbol{x}_i$ is the feature of the $i$-th sample before normalization and $K$ is the number of features with norm larger than $L$. In our experiments, $L = 1$ and the loss is directly added to the softmax loss

Table 1: ResNet50-D on ImageNet-1K classification dataset, under different logit formulations. The sparsity of validation features and the variability of training features are listed. FN denotes the feature norm regularization.

| Setting | Dim | Sparsity | Variability | | Accuracy | |
| --- | --- | --- | --- | --- | --- | --- |
| | | | Intra-Class | Inter-Class | Top1 | Top5 |
| Softmax | — | 0.55 | 60.12 | 90.01 | 78.04 | 93.89 |
| Cosine Softmax | — | 77.70 | 56.87 | 89.98 | 78.30 | 94.07 |
| Grassmannian | 1 | 77.92 | 56.52 | 89.98 | 78.48 | 94.24 |
| Grassmannian | 2 | 78.96 | 59.20 | 89.98 | 78.92 | 94.32 |
| Grassmannian | 4 | 78.49 | 61.25 | 89.99 | 78.86 | 94.34 |
| Grassmannian | 8 | 78.03 | 63.57 | 89.99 | 79.12 | 94.41 |
| Grassmannian | 16 | 78.12 | 65.81 | 90.00 | 79.21 | 94.29 |
| Grassmannian + FN | 1 | 33.51 | 56.32 | 90.02 | 78.65 | 94.24 |
| Grassmannian + FN | 8 | 45.55 | 63.01 | 90.02 | **79.37** | **94.53** |

with equal weight. We also tried larger values of $L$ or to regularize the norm of feature on both sides, however, they degrade the performance.

**Results**   The validation accuracies of different models on ImageNet-1K is listed in Table 1. All models with the Grassmannian class representation achieve higher top-1 and top-5 accuracies than the vanilla softmax and the cosine softmax. A general trend is that, with larger subspace dimension $k$, the accuracy improvement is greater. When subspace dimension is 16, the top-1 accuracy is 79.21%, which is 1.17% points higher than the vanilla softmax loss. With feature norm regularization, the top-1 accuracy further improves from 79.12% to 79.37% for dimension 8.

**Intra-Class Variability Increases with Dimension**   The intra-class variability is measured by the mean pair-wise angles (in degrees) between features within the same class, and then average over all classes. The inter-class variability is the average of mean pair-wise angles between features from different classes. Following the convention in the study of neural collapse (Papyan et al. (2020)), we use the global centered training feature to compute variabilities. Kornblith et al. (2021) showed that alternative objectives that improve accuracy, including label smoothing, dropout, sigmoid, cosine softmax, logit normalization, *etc.*, collapse the intra-class variability in representation, which in consequence degrades the quality of feature on downstream tasks. However, this conclusion does not apply when the classes are modeled by subspaces. The intra-class variability does reduces from baseline's 60.12 to Grassmannian formulation's 56.52 when the subspace dimension $k = 1$, however, *as $k$ increases, both the top-1 accuracy and the intra-class variability grow.* This indicates that representing classes as subspaces enables the simultaneous improvement of class discriminative power and expansion of intra-class variability.

**Feature Sparsity**   The feature sparsity is measured by the average percentage of zero activations on the validation set. As shown in Table 1, the feature from vanilla softmax networks are very dense, with only 0.55% zero activations. Cosine softmax and Grassmannian class representations all result in more sparse representations, with around 78% zero activations. The feature norm regularization decreases the sparsity about a half.

**Principal Angles Between Class Representative Spaces**   When classes are subspaces, relationships between two classes can be measured by $k$ angles called principal angles, which contain richer information than a single angle between two class vectors. The principal angles between two $k$-dimensional subspaces $S$ and $R$ are recursively defined as (Absil et al. (2006))

$$\cos(\theta_i) = \max_{\boldsymbol{s} \in S} \max_{\boldsymbol{r} \in R} \boldsymbol{s}^T \boldsymbol{r} = \boldsymbol{s}_i^T \boldsymbol{r}_i, s.t. \|\boldsymbol{s}\| = \|\boldsymbol{r}\| = 1, \boldsymbol{s}^T \boldsymbol{s}_j = \boldsymbol{r}^T \boldsymbol{r}_j = 0, j = 1, \dots, i - 1, \quad (13)$$

for $i = 1, \dots, k$ and $\theta_i \in [0, \pi/2]$. In Figure 2, we illustrate the smallest and largest principal angles between any pair of classes for a model with $k = 8$. From the figure, we can see that the smallest principal angle reflects class similarity, and the largest principal angle is around $\pi/2$. A smaller angle means the two classes are correlated in some directions, and a $\pi/2$ angle means that some directions in one class subspace is completely irrelevant (orthogonal) to the other class.

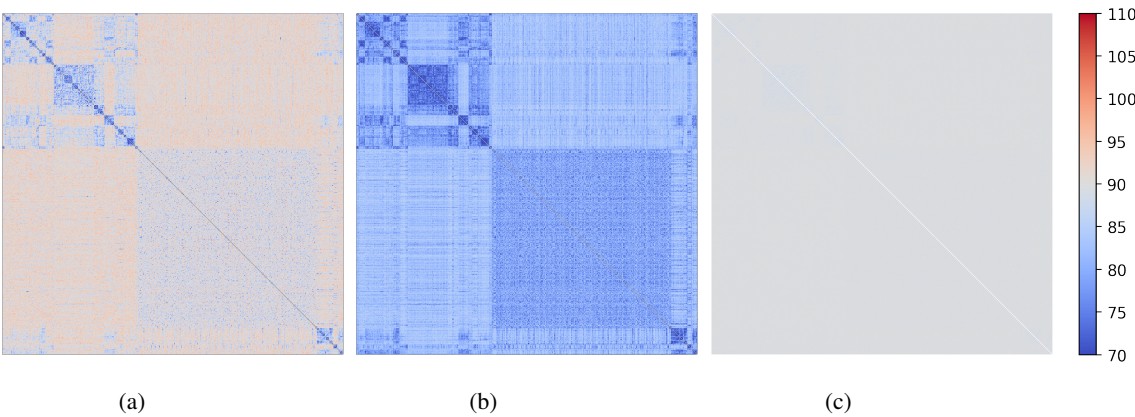

(a)          (b)          (c)

Figure 2: Each sub-figure is a heatmap of $1000 \times 1000$ grids. The color at the $i$-th row and the $j$-th column represent an angle between class $i$ and class $j$ in ImageNet-1K. (a) Pair-wise angles between class vectors of the ResNet50-D trained by vanilla softmax. Grids with red hue is large than $90°$, and blue hue means smaller than $90°$. (b) Pair-wise smallest principal angles between 8-dimensional class subspaces of a ResNet50-D model. Deeper blue colors indicates smaller angles. (c) Pair-wise largest principal angles of the same model as in (b). Grayish color means they are close to $90°$.

Table 2: Linear transfer using SVM. Models are the same as in Table 1. $R^2$ is the class separation.

| Setting | Dim | ImageNet | | Transfer | | | | | | |
|---|---|---|---|---|---|---|---|---|---|---|
| | | Top-1 | $R^2$ | CIFAR10 | CIFAR100 | Food | Pets | Cars | Flowers | Avg. |
| Softmax | — | 78.04 | 0.495 | 90.79 | 67.76 | 72.13 | 92.49 | 51.55 | 93.17 | 77.98 |
| Cosine Softmax | — | 78.30 | 0.528 | 89.34 | 65.32 | 64.79 | 91.68 | 43.92 | 87.28 | 73.72 |
| Grassmannian | 1 | 78.48 | 0.534 | 89.62 | 64.40 | 64.94 | 91.91 | 43.49 | 84.67 | 73.17 |
| Grassmannian | 2 | 78.92 | 0.493 | 89.71 | 66.33 | 66.08 | 91.44 | 45.11 | 87.81 | 74.41 |
| Grassmannian | 4 | 78.86 | 0.463 | 90.01 | 67.72 | 67.68 | 91.68 | 50.49 | 88.64 | 76.04 |
| Grassmannian | 8 | 79.12 | 0.429 | 90.63 | 68.25 | 69.85 | 91.83 | 51.83 | 94.06 | 77.74 |
| Grassmannian | 16 | 79.21 | 0.395 | 90.74 | 68.98 | 71.17 | 92.18 | 56.15 | 93.36 | 78.76 |
| Grassmannian + FN | 1 | 78.65 | 0.541 | 90.20 | 67.58 | 69.07 | 92.02 | 53.45 | 94.49 | 77.80 |
| Grassmannian + FN | 8 | **79.37** | 0.439 | **91.31** | **69.67** | **72.46** | **92.59** | **58.91** | **95.77** | **80.12** |

## 5.2 GRASSMANNIAN CLASS REPRESENTATION IMPROVES FEATURE TRANSFERABILITY

In this section we compare the linear transferability of the features learned by different models trained on the ImageNet-1K dataset. The feature transfer benchmark dataset includes CIFAR-10 (Krizhevsky et al. (2009)), CIFAR-100 (Krizhevsky et al. (2009)), Food-101 (Bossard et al. (2014)), Oxford-IIIT Pets (Parkhi et al. (2012)), Stanford Cars (Krause et al. (2013)), and Oxford 102 Flowers (Nilsback & Zisserman (2008)). For each of the transfer dataset, we use the same trained models as in Table 1 to extract their features. Then all features are normalized to unit length. We fit linear SVM with one-vs-rest multi-class policy on the training set, and report the accuracies on their test set. The regularization hyper-parameter for SVM is grid searched with candidates $[0.1, 0.2, 0.5, 1, 2, 5, 10, 15, 20]$ and determined by five-fold cross-validation on the training set.

**Results** As shown in Table 2, the cosine softmax and the Grassmannian with subspace dimension $k = 1$ has comparable transfer performance, but both are lower than the vanilla softmax. However, when the subspace dimension increases, the transfer performance gradually improves, and when $k = 16$, the transfer performance is on par with vanilla softmax. The feature norm regularization improves the transfer quality, as shown in the $k = 1, 8$ cases. We hypothesize that this might relate to the fact that features with norm regularization are less sparse, so more information are encoded.

**Class Separation** The class separation is measured by the index $R^2$, which is defined as one minus the ratio of the average intra-class cosine distance to the overall average cosine distance (Kornblith et al., 2021, Eq. (11)). Kornblith et al. (2021) found that greater class separation $R^2$ is associated with less transferable features. This may explain the feature transfer performance of Grassmannian class

Table 3: Long-tail classification on ImageNet-LT. Test accuracies are shown.

| Base Model | Dim | Many($> 100$) | Medium(20–100) | Few($< 20$) | Overall |
|---|---|---|---|---|---|
| Softmax | — | 65.42 | 40.64 | 15.97 | 46.83 |
| Cosine Softmax | — | 67.16 | 42.37 | 16.87 | 48.45 |
| Grassmannian | 1 | 67.70 | **42.87** | **17.22** | **48.94** |
| Grassmannian | 2 | 67.75 | 42.75 | 15.44 | 48.66 |
| Grassmannian | 4 | **68.02** | 40.82 | 12.88 | 47.49 |
| Grassmannian | 8 | 67.78 | 37.60 | 10.88 | 45.59 |

representations. The vanilla softmax has lower separation (0.495) compared to the cosine softmax (0.528) and the Grassmannian class representation with subspace dimension $k = 1$ (0.534). From subspace dimension $k = 1$ to $k = 16$, the separation from Grassmannian models decreases from a high value (0.534) to a low value (0.395). The change in class separation is roughly in line with the change of transfer performances.

### 5.3 Scaleless of Subspace Improves Long-tail Recognition

We benchmark its effectiveness in long-tail classification using the ImageNet-LT dataset (Liu et al. (2019)). ImageNet-LT is a subset of ImageNet-1K, where the number of images per class ranges from 5 to 1280. There are totally 115.8K images, roughly 1/10 the size of ImageNet-1K. We use the same ResNet50-D networks as in Section 5.1. All training settings including optimizer, augmentation, initial learning rate are also kept the same except we modify the total epochs to 200 and the learning rate is decayed by 1/10 at epoch 150, 180, and 195. The last checkpoint is used for evaluation. We use the instance-balanced sampling, as it was reported by Kang et al. (2019) that class-balanced sampling, and square-root sampling both degrade the performance.

We report the top-1 accuracies on the test set in Table 3. We find that both the cosine softmax and the Grassmannian class representation with small subspace dimension improve the long-tail classification accuracy. Specifically, the cosine softmax is 1.62% higher in score compared to the vanilla softmax, and the Grassmannian class representation with subspace dimension $k = 1$ is 2.11% higher in score compared to the vanilla softmax. However, when the subspace dimension increases, the accuracy drops. We notice that for few-shot classes, there are not enough sample to learn a good higher dimensional subspace for its representation, as the accuracy of few-shot classes degrade significantly when dimension are large. Too few training data for a class is an example scenario when larger dimension does not offer much help.

## 6 Limitation and Future Direction

One problem that remains open is how to choose the optimal dimension. Currently, we treat it as a hyper-parameter and decide it through experiments. Computational side, geometric optimization incurs some computational overhead since it contains SVD decomposition. This might hinder the training speed when $k$ is very large. The Grassmannian class representation allows for greater intra-class variability, but we did not explicitly promote the intra-class variability in any form. It will be very interesting to explore ways to explicitly encourage intra-class variability. For example, a potential way is to combine it with self-supervised learning. We hope our work would stimulate progress in these directions.

## 7 Conclusion

In this work, we proposed to use linear subspaces as the class prototype in deep neural networks. The geometric structure of the related Grassmannian fully-connected layer and the Grassmannian convolutional layer are products of Grassmannian. We optimize the subspaces using geometric optimization and provide an efficient Riemannian SGD implementation tailored for Grassmannians. We apply the new formulation to large-scale image classification, feature transfer, and long-tail classification tasks. Experiments demonstrate that the new Grassmannian class representation is able to improve performances in these settings.

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

Table 4: Validation accuracy of Grassmannian ResNet50-D on ImageNet with different retractions. The first two rows are two implementations of Riemannian SGD, while the third row is the vanilla SGD without any manifold constraints.

| Setting | Dim | Optimization | Retraction | Top1 | Top5 |
|---------|-----|--------------|------------|------|------|
| Grassmannian | 8 | RSGD Algorithm 1 | Move along geodesic | **79.12** | **94.41** |
| Grassmannian | 8 | RSGD Algorithm 1 variant | Retraction via QR | **79.13** | **94.45** |
| Unconstrained | 8 | SGD | None | 78.55 | 94.18 |

Saining Xie, Ross Girshick, Piotr Dollár, Zhuowen Tu, and Kaiming He. Aggregated residual transformations for deep neural networks. In *Proceedings of the IEEE conference on computer vision and pattern recognition*, pp. 1492–1500, 2017b.

Yaodong Yu, Kwan Ho Ryan Chan, Chong You, Chaobing Song, and Yi Ma. Learning diverse and discriminative representations via the principle of maximal coding rate reduction. *Advances in Neural Information Processing Systems*, 33:9422–9434, 2020.

Tong Zhang, Pan Ji, Mehrtash Harandi, Richard Hartley, and Ian Reid. Scalable deep k-subspace clustering. In *Asian Conference on Computer Vision*, pp. 466–481. Springer, 2018.

Xu Zhang, Felix X Yu, Sanjiv Kumar, and Shih-Fu Chang. Learning spread-out local feature descriptors. In *Proceedings of the IEEE international conference on computer vision*, pp. 4595–4603, 2017.

# A  TECHNICAL DETAILS

**Alternative Implementation of Riemannian SGD**   The step 4 of Algorithm 1 is called *retraction* in geometric optimization. There are alternative implementations of retraction other than moving parameters along the geodesic. For example, replace step 4 with the Euclidean gradient update and followed by the re-orthogonalization via QR decomposition in Step 5. The subspace parameter may move away from the Grassmannian after the Euclidean gradient update, but it will be pulled back to the manifold after the QR re-orthogonalization (details see Absil et al. (2009, Equ. (4.11))). For ease of reference, we call this version of Riemannian SGD as "Algorithm 1 variant". We compare the two implementations in the first two rows of Table 4. The results show that the Grassmannian class representation is effective on both versions of Riemannian SGD.

**Necessity of Grassmannian Formulation and Geometric Optimization**   To show that the necessity of constraining the subspace parameters to lie in the Grassmannian, we replace the Riemannian SGD with the vanilla SGD and compare it with Riemannian SGD. Note that with SGD, the logit formula $\|S_i^T x\|$ no longer means the projection norm because $S_i$ is not orthogonal anymore. The result is shown at the third row of Table 4, from which we observe a significant performance drop for the unconstrained setting.

**Numerical Stability of Algorithm 1**   The numerical stability issue is caused by the accumulation of tiny computational errors of Equation (3). After many iterations, the resultant matrix $S$ might not be perfectly orthogonal. For example, after 100, 1000, and 5000 iterations of the Grassmannian ResNet50-D with subspace dimension $k = 8$, we observed that the error $\max_i \|S_i^T S_i - I\|_\infty$ is 1.9e-5, 9.6e-5 and 3.7e-4, respectively. After 50 epochs, the error accumulates to 0.0075. One can run step 5 every 100 iterations to keep the error at low level and the computational cost is neglectable. For this reason, we marked this step as "optional".

**Decreasing Feature Norm During Training**   We show the changes of average norm on the validation set of ImageNet from epoch 10 to epoch 100 in Figure 3. The subspace dimension $k = 16$.

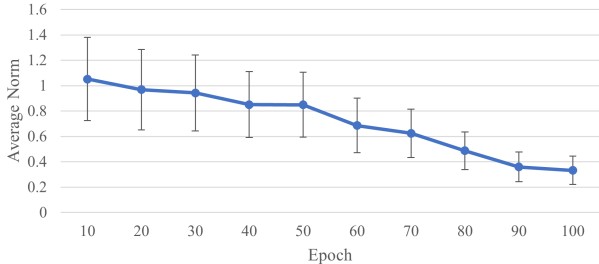

Figure 3: The average norm of features on the validation set. The error bar shows the std of norms.

Table 5: Validation accuracy of Grassmannian ResNet50-D on ImageNet with varying $\gamma$. Different from other experiments, this set of comparison was conducted using the step learning schedule.

| Setting | Dim | Lr Schedule | $\gamma$ | Top1 | Top5 |
|---|---|---|---|---|---|
| Grassmannian | 8 | Step | 16 | 78.27 | 93.87 |
| Grassmannian | 8 | Step | 20 | 78.55 | 94.05 |
| Grassmannian | 8 | Step | 25 | **78.71** | **94.11** |
| Grassmannian | 8 | Step | 32 | 78.47 | 94.07 |

## B   HYPER-PARAMETERS AND DESIGN DECISIONS

**Choice of Gamma**   We use $\gamma = 25$ throughout the main text. Here we give more results with different choice of $\gamma$ when subspace dimension $k = 8$ in Table 5. Due to that we conducted this set of experiments in early exploration stage, the learning rate decay policy is to divide by 10 at epochs 30, 60 and 90, which is different from our main results using the cosine learning rate schedule. The top-1 accuracy is slightly lower than the cosine learning rate counter part. Other training settings such as augmentation are the same as in Table 1.

**Importance of Re-Normalizing Features**   Re-normalizing the feature is critical to effectively learn the class representative subspaces. Below we provide training results without feature re-normalization in Table 6. There are significant performance drop without re-normalization. For reference, the cosine softmax also requires feature re-normalization for effective learning.

**Importance of Joint Training**   Joint training the subspaces and the features is essential. To support this claim, we add an experiment that only fine-tunes the class subspaces from weights pre-trained using the regular softmax (third row of Table 7). For comparison, we also add another experiment that fine-tunes all parameters (fourth row of Table 7). We find that if the feature is fixed, changing the regular fc to the geometric version does not increase performance noticeably (top-1 from 78.04% to 78.14%). But when all parameters are free to learn, the pre-trained weights is a better initialization than the random initialization (top-1 from 79.12% to 79.44%).

**More Results of FN**   We present more results using the feature norm regularization trick in Table 8. From the results, we observe that FN also works for the baseline Cosine Softmax. For Grassmannian + FN, the performance reaches peak at dimension $k = 8$ and then decreases when $k = 16$.

**Stronger Augmentation Improves Accuracy**   Generally speaking, stronger augmentation mitigates the overfitting problem and benefits models with larger capacity. To demonstrate the effect of stronger augmentations, we run experiments using RandAug (Cubuk et al. (2020)) in Table 9. We can see that stronger augmentation indeed further increases the accuracy. Together with longer training and SyncBN, the top-1 accuracy for ResNet50-D reaches 80.17%.

Table 6: Compare the validation accuracy of Grassmannian ResNet50-D on ImageNet with/without re-normalization.

| Setting | Dim | Re-Normalization | Top1 | Top5 |
|---------|-----|------------------|------|------|
| Grassmannian | 1 | ✓ | **78.48** | **94.24** |
| Grassmannian | 1 | | 77.91 | 93.78 |
| Grassmannian | 8 | ✓ | **79.12** | **94.41** |
| Grassmannian | 8 | | 78.12 | 93.90 |

Table 7: Validation accuracy of ResNet50-D on ImageNet trained with fine-tuning. Models of the last two rows are initialized using the weights from the first row.

| Setting | Dim | Initialization | Fine-Tune | Top1 | Top5 |
|---------|-----|----------------|-----------|------|------|
| Softmax | — | Random | — | 78.04 | 93.89 |
| Grassmann | 8 | Random | — | 79.12 | 94.41 |
| Grassmann | 8 | Weights trained using softmax | Only last layer | 78.14 | 93.97 |
| Grassmann | 8 | Weights trained using softmax | All parameters | **79.44** | **94.58** |

## C    MORE BASELINES

We have compared the proposed method with vanilla softmax and the cosine softmax the main text. In this section we compare with baselines that use the same amount of parameters, and run experiments on different network structures.

**Multi-FC**    We add multiple classification fc layer to the network. During training, these independent fcs are trained side by side, and their losses are averaged. During testing, the logits are first averaged, and then followed by softmax to output the prediction probability.

**SoftTriple**    In the SoftTriple loss (Qian et al. (2019)), each class is modeled by multiple centers. The logit is a weighted average of logits computed from individual class centers. We adapted the official code into our codebase to train on the ImageNet dataset. The recommended parameters are used. Specifically, $\lambda = 20, \gamma = 0.1, \tau = 0.2$ and $\delta = 0.01$.

For the above two settings, we use the same training protocols as in Table 1. Results are shown in Table 10, from which we find that the Grassmannian class representation is the most effective one.

**More Architectures**    We show experiments on ResNet101-D and ResNeXt (Xie et al. (2017b)) in Table 11. The training settings are the same as in Table 1, namely, we use the standard augmentation, cosine learning rate schedule, and train for 100 epochs. The results show that our formulation is effective across different model architectures.

## D    TRAINING SPEED AND SVD SPEED

During inference, the computational cost is $K$ times the vanilla softmax. Since it is mostly matrix multiplication, the GPU acceleration can speed up even further. For example, on a V100 GPU, the average time of multiplying a $1000 \times 2048$ matrix with a 2048 dimensional vector is $20 \pm 2.9 \mu s$, while multiplying an $8000 \times 2048$ matrix with a 2048 dimensional vector takes about $105 \pm 7.6 \mu s$. The cost is neglectable compared to the network forward time.

During training, the most costly operation in Algorithm 1 is SVD. We measure the actual iteration time during training in Table 12. We observe that when $K$ is small, it is as fast as the vanilla softmax. When $k = 8$, the full training needs roughly 1.7x time compared to vanilla softmax (this can be reduced greatly with the new version of PyTorch, as we will discuss below).

Since the release of PyTorch 1.13, they supported the fast approximate SVD algorithm GESVDA. We saw great speed improvement in the case of $k = 8$ and $k = 16$. The benchmark time is shown

Table 8: Validation accuracy of ResNet50-D on ImageNet with FN regularization.

| Setting | Dim | Top1 | Top5 |
|---|---|---|---|
| Cosine Softmax + FN | — | 78.64 | 94.24 |
| Grassmannian + FN | 1 | 78.65 | 94.24 |
| Grassmannian + FN | 2 | 78.87 | 94.43 |
| Grassmannian + FN | 4 | 79.10 | 94.58 |
| Grassmannian + FN | 8 | **79.37** | **94.53** |
| Grassmannian + FN | 16 | 79.09 | 94.37 |

Table 9: Validation accuracy of ResNet50-D on ImageNet with stronger augmentation and longer training. SyncBN also improves accuracy.

| Setting | Dim | Augmentation | SyncBN | Epochs | Top1 | Top5 |
|---|---|---|---|---|---|---|
| Softmax | — | Standard | | 100 | 78.04 | 93.89 |
| Softmax | — | RandAug | ✓ | 100 | 78.04 | 94.05 |
| Cosine Softmax | — | Standard | | 100 | 78.30 | 94.07 |
| Cosine Softmax | — | RandAug | ✓ | 100 | 78.95 | 94.55 |
| Grassmannian | 8 | Standard | | 100 | 79.12 | 94.41 |
| Grassmannian | 8 | RandAug | | 100 | 79.32 | 94.46 |
| Grassmannian | 8 | RandAug | ✓ | 100 | 79.49 | 94.64 |
| Grassmannian | 8 | RandAug | | 300 | 79.91 | 94.87 |
| Grassmannian | 8 | RandAug | ✓ | 300 | 80.03 | 94.77 |
| Grassmannian | 16 | Standard | | 100 | 79.21 | 94.29 |
| Grassmannian | 16 | RandAug | ✓ | 100 | 79.53 | 94.58 |
| Grassmannian | 16 | RandAug | ✓ | 300 | **80.17** | **94.93** |

in Table 13. With computational optimizations as such, we expect the computational cost of SVD would be minimal for $k \leq 32$.

## E   PYTORCH CODE FOR RIEMANNIAN SGD

We provide a sample implementation of Algorithm 1 in Figure 4 using PyTorch (Paszke et al. (2019)). The sample code checks if a parameter is geometric by checking whether it has an 'geometry' attribute. If not, then it runs the original SGD on that parameter. If the 'geometry' property is not `None`, then it is a list of numbers indicating the dimension of class representative subspaces for all classes. If all the dimensions are the same, then it goes to the batch version (line 23 of the code in Figure 4). Otherwise, it goes to the for loop version (line 46 of the code in Figure 4).

Table 10: Validation accuracy of ResNet50-D on ImageNet trained with Grassmannian, Multi-FC and SoftTriple. They use the same amount of parameters to represent a class.

| Setting | Class Parameters | Top1 | Top5 |
|---------|------------------|------|------|
| Grassmannian | $k = 8$ dimensional subspaces | **79.12** | **94.41** |
| Multi-FC | 8 fc classification layers | 77.34 | 93.65 |
| SoftTriple | 8 centers for each class | 75.55 | 92.62 |

Table 11: Validation accuracy of ResNet101-D and ResNeXt50 on ImageNet.

| Architecture | Setting | Dim | Top1 | Top5 |
|--------------|---------|-----|------|------|
| ResNeXt50 | Softmax | — | 78.02 | 93.98 |
| ResNeXt50 | Grassmannian | 8 | **79.00** | **94.28** |
| ResNet101-D | Softmax | — | 79.32 | 94.62 |
| ResNet101-D | Grassmannian | 8 | **80.03** | **94.81** |

Table 12: Average iteration time (forward + backward) during training on 8xV100 server. The PyTorch version used in this table is 1.11.0. The SVD step is computed on CPU due to PyTorch performance regression prior to version 1.13.0.

| Setting | Dim | Avg. Iter Time (ms) |
|---------|-----|---------------------|
| Softmax | — | 147 |
| Cosine Softmax | — | 150 |
| Grassmann | 1 | 149 |
| Grassmann | 2 | 145 |
| Grassmann | 4 | 177 |
| Grassmann | 8 | 256 |
| Grassmann | 16 | 449 |

Table 13: SVD time with Approximation on Nvidia GeForce GTX 1080 Ti using PyTorch 1.13.0. The tested matrix has shape (num classes, feature dimension, subspace dimension). The numbers should be smaller when running on more powerful devices such as V100.

| Matrix Size | CPU (ms) | GPU (ms) | GPU+GESVDA (ms) |
|-------------|----------|----------|------------------|
| $1000 \times 2048 \times 1$ | 3.1 | 117.7 | 41.3 |
| $1000 \times 2048 \times 2$ | 8.2 | 196.8 | 41.1 |
| $1000 \times 2048 \times 4$ | 18.9 | 366.3 | 41.8 |
| $1000 \times 2048 \times 8$ | 80.5 | 495.1 | 45.2 |
| $1000 \times 2048 \times 16$ | 211.9 | 627.1 | 55.4 |
| $1000 \times 2048 \times 32$ | 640.3 | 834.9 | 92.5 |

```python
 1  def _single_tensor_rsgd(params: List[Tensor], d_p_list: List[Tensor], momentum_buffer_list: List[Optional[Tensor]], *,
 2                          weight_decay: float, momentum: float, lr: float, dampening: float, maximize: bool):
 3
 4      for i, param in enumerate(params):
 5          d_p = d_p_list[i]
 6          if weight_decay != 0 and not hasattr(param, 'geometry'):
 7              d_p = d_p.add(param, alpha=weight_decay)
 8          if momentum != 0:
 9              buf = momentum_buffer_list[i]
10              if buf is None:
11                  buf = torch.clone(d_p).detach()
12                  momentum_buffer_list[i] = buf
13              else:
14                  # new momentum (step 3 in Alg. 1)
15                  buf.mul_(momentum).add_(d_p, alpha=1 - dampening)
16              d_p = buf
17
18          alpha = lr if maximize else -lr
19          buf = momentum_buffer_list[i]
20
21          if hasattr(param, 'geometry'):
22              # geometric SGD for parameters with 'geometry' attribute
23              if len(np.unique(param.geometry)) == 1:
24                  # batch version
25                  subdim = param.geometry[0]
26                  featdim = param.size(1)
27                  # reshape
28                  batch_p = param.reshape(-1, subdim, featdim).permute([0, 2, 1])
29                  batch_d_p = d_p.reshape(-1, subdim, featdim).permute([0, 2, 1])
30                  batch_m = buf.reshape(-1, subdim, featdim).permute([0, 2, 1])
31                  # euclidean grad to riemannian grad (equation (10) and (13))
32                  batch_g = batch_d_p - batch_p @ (batch_p.permute([0, 2, 1]) @ batch_d_p)
33                  # update momentum
34                  buf.add_(batch_g.permute([0, 2, 1]).reshape(-1, featdim) - buf)
35                  # svd
36                  (batch_U, batch_s, batch_Vt) = torch.linalg.svd(batch_g, full_matrices=False)
37                  # new parameter by moving along geodesic
38                  new_batch_p = (
39                      (batch_p @ batch_Vt.permute([0, 2, 1])) * torch.cos(alpha * batch_s).reshape(-1, 1, subdim) +
40                      batch_U * torch.sin(alpha * batch_s).reshape(-1, 1, subdim)) @ batch_Vt
41                  # qr decomposition
42                  new_batch_p = torch.linalg.qr(new_batch_p).Q
43                  # update param
44                  param.add_(new_batch_p.permute([0, 2, 1]).reshape(-1, featdim) - param)
45              else:
46                  # general case
47                  dims = [0, *np.cumsum(param.geometry)]
48                  for c, cc in zip(dims[:-1], dims[1:]):
49                      # reshape
50                      p = param[c:cc].T
51                      d = d_p[c:cc].T
52                      m = buf[c:cc].T
53                      # euclidean grad to riemannian grad (equation (10) and (13))
54                      g = d - p @ (p.T @ d)
55                      # update momentum
56                      m.add_(g - m)
57                      # svd
58                      (U, s, Vt) = torch.linalg.svd(g, full_matrices=False)
59                      # new parameter by moving along geodesic
60                      new_p = torch.cat((p @ Vt.T, U), 1) @ torch.cat(
61                          (torch.diagflat(torch.cos(alpha * s)), torch.diagflat(torch.sin(alpha * s))), 0) @ Vt
62                      # qr decomposition
63                      new_p = torch.linalg.qr(new_p).Q
64                      # update param
65                      p.add_(new_p - p)
66          else:
67              # original SGD
68              param.add_(d_p, alpha=alpha)
69
```

Figure 4: Pytorch implementation of the Riemannian SGD of Algorithm 1. The function shown in this figure is the core part. To make it runnable, one can copy an implementation of SGD (https://github.com/pytorch/pytorch/blob/master/torch/optim/sgd.py), and replace the _single_tensor_sgd() function with _single_tensor_rsgd() shown above.

