# OpenReview forum: "Grassmannian Class Representation in Deep Learning"
_ICLR.cc/2023/Conference — Submitted to ICLR 2023_

### Official Review · Reviewer_Acxf · 2022-10-23

**Confidence:** 5
**Correctness:** 2
**Technical Novelty And Significance:** 2
**Empirical Novelty And Significance:** 2
**Recommendation:** 5

**Clarity, Quality, Novelty And Reproducibility:**

The novelty of this work is considered marginal with some reasons explained in the weaknesses, the clarity of the statement needs to be improved, and currently there is no code for reproducibility.



**Strength And Weaknesses:**

Strengths:
- The feature norm regularization is somewhat novel for training the neural network under geometry constraints.
- This work has a strong performance in term of empirical results compared to prior methods in image classification and transfer learning.
- The empirical results also show that the representations using the Grassmannian layer is more sparse.

Weaknesses :
- This work has unclear motivation why second order representations in the form of linear subspaces yields better performance compared to the first order representations.  There is no motivating examples nor theories when subspaces are suitable representing classes. Citing the work Watanabe and Pakvasa does not directly describe why the linear subspace approach is a better model to represent classes in the era of deep learning.
- The novelty of this work is marginal with many overlapping points and contributions compared to the work of Simon et al. The problems of image classification and transfer learning are covered by the work of Simon et al. that enjoys the superiority of linear subspaces over prototypes (a single vector) to represent classes in few-shot learning. In experiments,  this work does not discuss or even compare with the proposed method.
- The proposed method updaates linear subspaces in classifiers with some constraints, and also this is not novel as some other works by Harandi and Fernando “Generalized BackPropagation Etude De Cas: Orthogonality” and Roy et al., “Siamese Networks: The Tale of Two Manifolds” have discussed similar concepts (i.e. the geometry aware layers) and implemented the proposed method for image classification but this works has no comparison to these prior works.
- Moreover, the types of data feasibly represented using the linear subspace method are also not discussed in the paper. Is the proposed method only applicable for visual data?
- This statement “The whole matrix S needs not be orthonormal”. For discriminative purposes, even though it needs more investigation, the straightforward idea is to force subspaces as different as possible to avoid collapses (see Simon et al., Arjovsky et al., Ozay and Okatani, and Harandi et al.). However, this work does not have any discussion about this idea nor include the idea to discriminate between subspaces.
- There are no properties of the proposed Grassmannian class representation layers.  For instance, what are the properties and benefits preserving the orthogonality for each subspace? what are properties of not preserving subspaces coming from different to be orthogonal?
- The design of this approach is somehow limited for a neural network module. How is the design of the proposed method with the multi-layer version of fully-connected layers (if possible)?
- The experiments require some prior methods for comparison with some variants in class representations, e.g., prototypes (the average of all representations within a class), non-learnable subspaces (a similar concept as in Simon et al.), .
- The performance of long-tail classification is marginally improved compared to cosine softmax.  That shows that the proposed method might not be quite effective in addressing such issue compared to transfer learning and common image classification.
- Is there any comparison in terms of speed between softmax strategies and the Grassmanian one? The discussion of trade-off between the performance gain and the processing time is crucial for this type of method because it usually requires additional processing time especially with contstraint optimization.
- The experiments are also lacking of comparison to some other models as a backbone. For instance, the proposed method can compare the methods using transformer models, another ResNet type (e.g., ResNet101), VGG, Inception.
- The feature sparsity is not very clear, is that 78% zero activations on the feature before the Grasmannian layer? or the elements of the Grassmannian layer (i.e., each subspace)? or the output after the Grassmannian layer?



References:

Simon et al, “Adaptive Subspaces for Few-Shot Learning,“ CVPR, 2020.

Harandi et al., “Extrinsic Methods for Coding and Dictionary Learning on Grassmann Manifolds,” IJCV, 2015.

Arjovsky et al. "Unitary evolution recurrent neural networks," ICML, 2016.

Ozay and Okatani, "Training CNNs with normalized kernels," AAAI, 2018.

**Summary Of The Paper:**

This paper proposes a method to represent a class as a subspace in the deep learning regime. The contributions of this paper are the formulation of classes as subspaces, the Grassmannian layer to update subspaces, learning the Grassmannian layer using constraint optimization. The core contribution is to represent a class as a subspace where we can easily replace the softmax layer with fully-connected networks with the projection between subsapaces and the input feature. The experiments show strong performance improvement on some cases on large datasets (e.g., ImageNet). As an example of its effectiveness compared to the softmax one, the top-1 accuracy of ImageNet1K is improved by 1.3%.


**Summary Of The Review:**

This paper has some insights about the use of linear subspaces for image classification especially comparing with the vanilla fully-connected layer in neural networks. However, there are some issues regarding novelty, comparison, and experiments.

---

> ### Author Response · Authors · 2022-11-16
> **Response to R5 (Part 3)**
>
> ### Q10
>
> > Is there any comparison in terms of speed between softmax strategies and the Grassmannian one? The discussion of trade-off between the performance gain and the processing time is crucial for this type of method because it usually requires additional processing time especially with constraint optimization.
>
> Thanks for the good question.
> During inference, the computational cost is K times the vanilla softmax.
> Since it is mostly matrix multiplication, the GPU acceleration can speed up even faster.
> For example, on a V100 GPU, the average time of multiplying a 1000x2048 matrix with a 2048 dimensional vector is 20±2.9μs, while multiplying an 8000x2048 matrix with a 2048 dimensional vector takes about 105±7.6μs.
> The cost is neglectable compared to the network forward time.
>
> During training, the most costly operation is SVD.
> We measure the actual iteration time in training in Tab. 5-2.
> We observe that when K is small, it is as fast as the vanilla softmax.
> When K=8, the full training needs roughly 1.7x time compared to vanilla softmax (this can be reduced greatly with the new version of PyTorch, as we will discuss below).
>
> Table 5-2. Average iteration time (forward + backward) during training on 8xV100 server. The SVD step is computed on CPU due to PyTorch performance regression prior to version 1.13.0.
>
> | Setting        | Dim | Avg. Iter Time (ms) |
> | -------------- | --- | :-----------------: |
> | Softmax        | -   |         147         |
> | Cosine Softmax | -   |         150         |
> | Grassmann      | 1   |         149         |
> | Grassmann      | 2   |         145         |
> | Grassmann      | 4   |         177         |
> | Grassmann      | 8   |         256         |
> | Grassmann      | 16  |         449         |
>
> Since the release of PyTorch 1.13 on Oct 29, 2022, about 19 days ago, they supported the fast approximate SVD algorithm GESVDA.
> We saw great speed improvement when K=8 and 16.
> The benchmark time is shown in Tab. 5-3.
> With optimizations such as this, we expect the computational cost would be minimal.
>
> Table 5-3. SVD Time (ms) with Approximation on Nvidia GeForce GTX 1080 Ti. The numbers should be smaller when running on more powerful devices such as V100.
>
> | Matrix Size      | CPU (ms) | GPU (ms) | GPU+GESVDA (ms) |
> | ---------------- | -------: | -------: | --------------: |
> | 1000 x 2048 x 1  |      3.1 |    117.7 |            41.3 |
> | 1000 x 2048 x 2  |      8.2 |    196.8 |            41.1 |
> | 1000 x 2048 x 4  |     18.9 |    366.3 |            41.8 |
> | 1000 x 2048 x 8  |     80.5 |    495.1 |            45.2 |
> | 1000 x 2048 x 16 |    211.9 |    627.1 |            55.4 |
> | 1000 x 2048 x 32 |    640.3 |    834.9 |            92.5 |
>
> ### Q11
>
> > The experiments are also lacking of comparison to some other models as a backbone. For instance, the proposed method can compare the methods using transformer models, another ResNet type (e.g., ResNet101), VGG, Inception.
>
> Thanks for pointing out this issue.
> We add more experiments in Tab. 5-4, including results of ResNet101, and another architecture ResNeXt (Xie et al. (2017)).
> The training settings are the same as in Table 1, namely, we use the standard augmentation, cosine learning rate schedule, and training for 100 epochs.
> The results show that our formulation is effective on these models.
>
> Table 5-4. Grassmannian with Dim=8 on ImageNet with different network structures
>
> | Architecture  | Setting      | Dim | Top-1 | Top-5 |
> | ------------- | ------------ | --- | :---: | ----: |
> | ResNet101-V1D | Softmax      | -   | 79.32 | 94.62 |
> | ResNet101-V1D | Grassmannian | 8   | 80.03 | 94.81 |
> | ResNeXt50     | Softmax      | -   | 78.02 | 93.98 |
> | ResNeXt50     | Grassmannian | 8   | 79.00 | 94.28 |
>
> ### Q12
>
> > The feature sparsity is not very clear, is that 78% zero activations on the feature before the Grassmannian layer? or the elements of the Grassmannian layer (i.e., each subspace)? or the output after the Grassmannian layer?
>
> Thanks for the question.
> The sparsity is the activations of features before the Grassmannian layer (so we used the term "feature sparsity" in the manuscript).
> The measurement of feature sparsity follows the work of Kornblith et al. (2021).
>
> ### References
>
> - Papyan, Vardan, X. Y. Han, and David L. Donoho. "Prevalence of neural collapse during the terminal phase of deep learning training." Proceedings of the National Academy of Sciences 117.40 (2020): 24652-24663.
> - Xie, Saining, et al. "Aggregated residual transformations for deep neural networks." Proceedings of the IEEE conference on computer vision and pattern recognition. 2017.
> - Kornblith, Simon, et al. "Why do better loss functions lead to less transferable features?." Advances in Neural Information Processing Systems 34 (2021): 28648-28662.

---

> ### Author Response · Authors · 2022-11-16
> **Response to R5 (Part 2)**
>
> ### Q5
>
> > This statement “The whole matrix S needs not be orthonormal”. For discriminative purposes, even though it needs more investigation, the straightforward idea is to force subspaces as different as possible to avoid collapses (see Simon et al., Arjovsky et al., Ozay and Okatani, and Harandi et al.). However, this work does not have any discussion about this idea nor include the idea to discriminate between subspaces.
>
> Thanks for the comment.
> The discrimination between subspaces is not an issue for our formulation for the following reasons.
>
> 1. The softmax loss itself can avoid collapse. Details can be found in the study of neural collapse (Papyan et al. (2020)) which we referenced in the introduction section.
>    Theoretically, the feature learned by the softmax forms an Equiangular Tight Frame (ETF), where classes are maximally separated.
>    Experimentally, we did not observe any collapse.
>    See Figure 2 in the manuscript for the learned angles between classes.
> 2. The referenced work of Arjovsky et al., Ozay et al. use orthogonal constraints to tackle the vanishing and exploding gradients in training deep networks.
>    It is a different category of problems compared with the collapse of subspaces.
>    The vanishing and exploding gradients in deep learning have largely mitigated, if not solved, by techniques such as BN and residual network structure, and our experiment does not suffer from this issue.
>
> ### Q6
>
> > There are no properties of the proposed Grassmannian class representation layers. For instance, what are the properties and benefits preserving the orthogonality for each subspace? what are properties of not preserving subspaces coming from different to be orthogonal?
>
> Thanks for asking.
> We do not quite follow the question, we try our best to answer based on our understanding.
>
> The orthogonal constraint is a numerical instantiation of the geometric object "subspace" (details can be found in the preliminary section).
> The benefit is that it is computationally friendly so that we can actually do Riemannian SGD on the abstract Grassmannian manifold.
>
> The principal angles between subspaces need not be orthogonal (See Figure 2 in the manuscript for the visualization of learned angles).
> Intuitively, a smaller angle indicates more visual similarity between the two classes.
> Actually, if we have 1000 classes (as in the ImageNet dataset) and let the subspace dimension be 8, then mathematically they cannot be all orthogonal to each other in a 2048-dimensional feature space.
>
> ### Q7
>
> > The design of this approach is somehow limited for a neural network module. How is the design of the proposed method with the multi-layer version of fully-connected layers (if possible)?
>
> Thanks for the good question.
> The subspace formulation is tailored for the representation of classes and we do not intend to generalize it to intermediate layers of a deep network since there are yet no concrete concepts (such as classes) we want to model.
> Nevertheless, we want to comment that for geometric constraints on general deep model layers, one can refer to the study of orthogonal constraints, which has a different purpose from ours.
>
> ### Q8
>
> > The experiments require some prior methods for comparison with some variants in class representations, e.g., prototypes (the average of all representations within a class), and non-learnable subspaces (a similar concept as in Simon et al.)
>
> Prototypes are designed for few-shot learning where the data is severely limited, and it is not a proper baseline for large-scale classification tasks.
>
> To compare with non-learnable subspaces, we design an experiment as follows.
> First, we fix the parameters of a ResNet50-V1D network trained on ImageNet, and then we replace the last linear fc with our subspace formulation and fine-tune the subspace weights to find the best subspace describing the dataset.
> The results are shown in Tab 5-1, and we observe that subspaces without learning do not improve performance noticeably, and thus the end2end feature learning is critical.
>
> Table 5-1. Compare with non-learnable subspaces.
>
> | Setting                            | Initialization                       | Top-1 | Top-5 |
> | ---------------------------------- | ------------------------------------ | :---: | ----: |
> | Vanilla Softmax                    | Random                               | 78.04 | 93.89 |
> | Grassmann, Fine-Tune Only Subspace | Weights trained from vanilla softmax | 78.14 | 93.97 |
>
> ### Q9
>
> > The performance of long-tail classification is marginally improved compared to cosine softmax. That shows that the proposed method might not be quite effective in addressing such issue compared to transfer learning and common image classification.
>
> Thanks for the comment.
> We agree and as we analyzed in the main text, the unexciting performance in long-tail classification when subspace dimension is large might be caused by insufficient data for the tail classes.

---

> ### Author Response · Authors · 2022-11-16
> **Response to R5 (Part 1)**
>
> We would like to thank the reviewer for the detailed comments on the manuscript.
> Below we answer the questions.
>
> ### Q1
>
> > This work has unclear motivation why second order representations in the form of linear subspaces yields better performance compared to the first order representations. There is no motivating examples nor theories when subspaces are suitable representing classes. Citing the work Watanabe and Pakvasa does not directly describe why the linear subspace approach is a better model to represent classes in the era of deep learning.
>
> Thanks for raising the question.
> The citation is for an introductory historical review for the idea of representing classes as subspaces.
> Other than the similarity in the basic abstract idea, the work of Watanabe et al. is technically vastly different from ours in terms of the concrete formulation, the optimization, and the application.
> We cite it to **pose** the question that "whether representing classes as subspaces is better in deep learning" and we answer this core question using the whole presentation of the manuscript.
>
> ### Q2
>
> > The novelty of this work is marginal with many overlapping points and contributions compared to the work of Simon et al. The problems of image classification and transfer learning are covered by the work of Simon et al. that enjoys the superiority of linear subspaces over prototypes (a single vector) to represent classes in few-shot learning. In experiments, this work does not discuss or even compare with the proposed method.
>
> Thanks for the comment and will add the work of Simon et al. to related work, but we disagree with the evaluation on novelty.
> We summarize the major differences below.
>
> 1. Problem setting.
>    Simon et al. solve the few-shot learning problem, while we focus on the large-scale classification problem.
>    Both the training objective and the evaluation protocol are different.
> 2. Subspace formulation and optimization.
>    The subspace formulation is different in the two papers.
>    Simon et al. do not have parameters to represent the learned subspace, rather, they estimate the subspace online using the collected features in each iteration.
>    There is no need for geometric optimization since there are no parameters to be learned.
>    This is essentially different from ours where subspaces are represented by weights and learned using geometric optimization.
> 3. Training procedure.
>    In the training algorithm at Simon et al., the average of the class needs to be computed for each iteration.
>    This is only applicable to the few-shot scenario and the cost is simply intractable for large-scale datasets.
>    For example, inferencing the whole ImageNet (to compute the class center) needs 300s in an 8xGPU workstation, while the normal iteration time is 0.15s on the same machine.
>
> Due to the aforementioned reason, there are very few overlaps technically and it is not a proper baseline for our work.
>
> ### Q3
>
> > The proposed method updates linear subspaces in classifiers with some constraints, and also this is not as novel as some other works by Harandi and Fernando “Generalized BackPropagation Etude De Cas: Orthogonality” and Roy et al., “Siamese Networks: The Tale of Two Manifolds” have discussed similar concepts (i.e. the geometry aware layers) and implemented the proposed method for image classification but this works has no comparison to these prior works.
>
> Thanks for the comment and we will add the two works in the related work section.
> However, both are not proper baselines for our work.
> The work of Harandi et al. belongs to the category of "orthogonal constraints" which we have briefly reviewed in the related work section.
> The work of Roy et al. is designed for deep metric learning where they learned a Mahalanobis distance matrix using geometric optimization.
> Although they evaluated on classification dataset, their metrics used are NMI (commonly used to evaluate clustering techniques) and Recall@K (used to evaluate retrieval performance), instead of accuracy (for classification).
> Their paper did not compare with standard classification either.
>
> ### Q4
>
> > Moreover, the types of data feasibly represented using the linear subspace method are also not discussed in the paper. Is the proposed method only applicable for visual data?
>
> Thanks for the question.
> There is nothing special in the formulation that ties to the image domain.
> Our formulation can be a drop-in replacement for the normal logit for classification problems, regardless of the domain of the data.

---

> ### Comment · Reviewer_Acxf · 2022-12-05
> **Response to authors**
>
> I'd like to thank the authors for such long and complete response. All the answers that directly targets my questions and doubts are appreciated. The rebuttal has included 5 more thorough experiments to provide more analysis, comparison, and clear my doubts. Through the rebuttal, the concerns about novelty and experiments are easy to reconcile.  Indeed, this work has a valuable position as an additional tool for classification tasks using deep learning.
>
> Despite all the good points mentioned above, this work has to admit that the processing time is much slower in the forward and backward mode compared to the baselines. Moreover, this work has a lack of comparison to related works. For instance, this work can compare with the nearest neighbor classifier using the prototypical method (e.g., outputs/latents/features from a pretrained model). The proposed method can also be compared with the methods using orthogonal constraints e.g., some methods in Section 2 (Arjovsky
> et al. (2016); Xie et al. (2017a); Bansal et al. (2018); Qi et al. (2020); Wang et al. (2020), etc., Ozay & Okatani (2018); Harandi & Fernando (2016), and Su et al. (2022)). All baselines in the paper are in the basic form and do not reflect the advent methods in this research area.
>
> Considering all the benefits and drawbacks of the proposed approach, I'd like to increase the score.

---

> > ### Author Response · Authors · 2022-12-12
> > **Follow-up response to R5**
> >
> > Dear Reviewer Acxf,
> >
> > We would like to thank you for reading our response and sincerely appreciate that you have raised the score.
> > For the remaining concerns, we provide more data points and perspectives and hope to clarify misunderstandings (if any).
> >
> > > this work has to admit that the processing time is much slower in the forward and backward mode compared to the baselines.
> >
> > Thanks again for the question. In our response to Q10, we showed that the influence on forward time is neglectable.
> > To further support this statement, we measure the wall time of inferencing 50,000 validation images 5 times and report their average time for forwarding one image in Tab. 5-5.
> > In the experiment, the batch size is 32 and we used 1 V100 GPU and set up 8 parallel data-loading workers.
> >
> > Table 5-5. Average forward time for one image.
> >
> > | Setting      | Dim | Avg. Time per Img (ms) |    FLOPs    |   FC FLOPs   | FC FLOPs Percentage |
> > | ------------ | --- | ---------------------- | :---------: | :----------: | ------------------: |
> > | Softmax      | -   | 1.424 $\pm$ 0.043      | 4.37 GFLOPs | 0.002 GFLOPs |        0.047% FLOPs |
> > | Grassmannian | 8   | 1.408 $\pm$ 0.066      | 4.38 GFLOPs | 0.016 GFLOPs |        0.374% FLOPs |
> >
> > From Tab. 5-5 we did not observe any noticeable time difference in inference.
> > Furthermore, the computation (FLOPs) of FC contributes less than 1% of the network inference.
> >
> > For the forward+backward pass, we admit that there is at most about 40% computational overhead.
> > We have analyzed in the response to Q10 that the bottleneck is the computation of SVD.
> > With the GPU+GESVDA backend, the overhead of SVD is estimated to be less than 55ms, which accounts for 38% overhead compared to the vanilla softmax.
> > On the one hand, future improvements in SVD computation may bring the cost even lower.
> > On the other hand, future work can seek to speed up the algorithm.
> > For example, one possible direction is to accumulate the gradients of Grassmannian FC and update its parameters every two iterations, which will cut the cost in half.
> >
> > > this work can compare with the nearest neighbor classifier using the prototypical method (e.g., outputs/latents/features from a pretrained model).
> >
> > Thanks for the suggestion.
> > We implemented the prototypical method using a pre-trained ResNet50-V1D network which is trained by the mmclassification team at https://github.com/open-mmlab/mmclassification/tree/master/configs/resnet.
> > We randomly sample 200,000 training images and use their class-wise average feature as the prototype for each class.
> > When testing, we find the nearest prototype for each of the validation images.
> > The results are shown in Tab. 5-6 and we observe a huge performance drop compared to the original softmax layer, in terms of both top-1 and top-5 accuracies.
> > This is not surprising, since, for fixed deep features, the original softmax has learned the nearly optimal linear classifier.
> > The class centers obtained by prototypes are not supposed to be better.
> >
> > Table 5-6. Prototypical method on ImageNet with pre-trained ResNet-V1D.
> >
> > | Setting | Original Top-1 | Original Top-5 | Prototypical Top-1 | Prototypical Top-5 |
> > | ------- | -------------- | :------------: | :----------------: | -----------------: |
> > | Softmax | 77.54          |     93.57      |       63.20        |              85.16 |
> >
> > > The proposed method can also be compared with the methods using orthogonal constraints e.g., some methods in Section 2
> >
> > Thanks for bringing about this issue again.
> > We have explained in the response why they are not proper baselines of our work.
> > Here we provide another perspective to help the clarification.
> > Works of orthogonal constraints modify the entire backbone network.
> > Their modification happens before the last deep feature layer, and they do not change the definition of logits.
> > Our work does not change the backbone network.
> > We changed the formulation of logit, which happens after the last deep feature layer.
> > So works of orthogonal constraints are "orthogonal" to our work.

---

### Official Review · Reviewer_ShD7 · 2022-10-25

**Confidence:** 3
**Correctness:** 4
**Technical Novelty And Significance:** 4
**Empirical Novelty And Significance:** Not applicable
**Recommendation:** 6

**Clarity, Quality, Novelty And Reproducibility:**

Clarity: good.

Quality: fair.

Novelty: good.

Reproducibility: good.

**Strength And Weaknesses:**

Strength:

(1) Introducing subspace learning in deep neural networks is interesting, and introducing geometric optimization with Riemannian SGD is useful to solve this problem.

(2) Experimental results on three tasks show improvements over traditional softmax methods.

(3) The paper is well written and organized.

Weaknesses:

(1) With larger k, the proposed method introduces much more parameters, not to say the SVD operation in the Riemannian SGD solver. As in Table 2, without FN only k=16 shows improvements. This would make a very difficult scalability for large-scale learning, for example, with millions of classes in training face recognition models.

(2) If computation is not an issue, traditionally there are also methods in expanding the class representative vectors, e.g. multiple experts and fusion. It is not clear if the improvement is due to enlarged classification parameters or due to the new learning framework. Therefore, it would be better to show a comparison against multiple experts.

(3) The FN loss contributes a lot for the improvements. However, it should be a general trick that can also be applied on the traditional softmax baselines, which should also be reported for a fair comparison. Without the FN loss it appears that the improvement of the proposed method is still limited.

(4) With k=1 the linear subspaces degrade to class vectors. In such case what is the difference between the proposed method with k=1 and cosine softmax? They perform quite similar with each other across all the three tables.

(5) How will the proposed method incorporate margin parameters and what would be its effect?

**Summary Of The Paper:**

In this paper, traditional class representative vectors deep neural networks are replaced by linear subspaces on Grassmann manifolds. It is supposed to be more informative for intra-class feature variations. The proposed method optimizes the subspaces using geometric optimization, with an efficient Riemannian SGD implementation tailored for Grassmannians. Experiments on image classification, feature transfer, and long-tail classification tasks show that the new method improves the vanilla softmax and cosine softmax.

**Summary Of The Review:**

The proposed new formulation for learning linear subspaces in deep neural networks is very interesting. However, from my point of view this work is still not yet solid. As I can understand the proposed method is novel and I expect a lot from it. However, after reading the experimental results I'm not that excited, and I think the proposed method is not yet fully validated in some aspects, as listed in weaknesses. To me the only reason to accept the current paper is that I would like to encourage this novel study.

---

> ### Author Response · Authors · 2022-11-16
> **Response to R4 (Part 2)**
>
> ### Q4
>
> > With k=1 the linear subspaces degrade to class vectors. In such case what is the difference between the proposed method with k=1 and cosine softmax? They perform quite similar with each other across all the three tables.
>
> Thanks for the question.
> When K=1, the logit becomes $l_i = \frac{|\boldsymbol{w}_i^T \boldsymbol{x}|}{\lVert\boldsymbol{w}_i\rVert}$.
> It is very similar to the cosine softmax $l_i = \frac{\boldsymbol{w}_i^T \boldsymbol{x}}{\lVert\boldsymbol{w}_i\rVert}$, except that the cosine softmax is signed.
> From this point of view, the Cosine Softmax is the more appropriate baseline compared to vanilla Softmax.
>
> ### Q5
>
> > How will the proposed method incorporate margin parameters and what would be its effect?
>
> Thanks for the question.
> When classes are represented by subspaces, we can still compute the angle between the feature vector and the subspace, or the cosine of this angle.
> So adding margins is straightforward as we add margins to the Cosine Softmax.
> For example,
> To see the effect of adding margins to the classification accuracy, we add an experiment in Tab. 4-3, where we add CosFace-style margins.
> From the results, we observe a performance drop when adding margins.
> Perhaps the feature discrimination power enhanced by margins only works with applications like face recognition.
> For reference, we equip the original ResNet50-V1D with the ArcFace loss (we use the same angular margin 0.5 and feature scale 64 as in training faces) and train on ImageNet, the resultant top-1 and top-5 are 71.80% and 92.07% respectively.
> It is significantly lower than the vanilla softmax.
>
> Table 4-3. Compare Grassmannian ResNet50-V1D with K=8 on ImageNet with margins.
>
> | Setting            | Margin | Top-1 | Top-5 |
> | ------------------ | ------ | :---: | ----: |
> | Grassmannian Dim 8 | -      | 79.12 | 94.41 |
> | Grassmannian Dim 8 | 0.15   | 78.81 | 93.61 |
>
> ### References
>
> - An, Xiang, et al. "Partial fc: Training 10 million identities on a single machine." Proceedings of the IEEE/CVF International Conference on Computer Vision. 2021.
> - Kornblith, Simon, et al. "Why do better loss functions lead to less transferable features?." Advances in Neural Information Processing Systems 34 (2021): 28648-28662.
> - Wang, Hao, et al. "Cosface: Large margin cosine loss for deep face recognition." Proceedings of the IEEE conference on computer vision and pattern recognition. 2018.
> - Deng, Jiankang, et al. "Arcface: Additive angular margin loss for deep face recognition." Proceedings of the IEEE/CVF conference on computer vision and pattern recognition. 2019.

---

> ### Author Response · Authors · 2022-11-16
> **Response to R4 (Part 1)**
>
> We would like to thank the reviewer for the thoughtful review and comments on the manuscript.
>
> ### Q1
>
> > With larger k, the proposed method introduces much more parameters, not to say the SVD operation in the Riemannian SGD solver. As in Table 2, without FN only k=16 shows improvements. This would make a very difficult scalability for large-scale learning, for example, with millions of classes in training face recognition models.
>
> We thank the reviewer for raising the issue.
> For a very large number of classes such as face recognition, the vanilla version of fc is already too large to be tractable.
> Partial FC (An et al. (2021)) is a technique to solve this problem by distributing fc weights on different GPUs and sparsely updating the fc weights.
> Such techniques can also be applied to our framework so that both the storage issue and the computational issue could be solved.
>
> For Table 2, we mainly address the problem discovered by Kornblith et al. (2021), which states that the vanilla softmax has the best transferable features and all alternative losses, such as the Cosine Softmax, increase the classification accuracy by harming feature transferability.
> Our experiments in Table 2 show that, by increasing the subspace dimensions, the feature transferability is recovered and at the same time, the top-1 accuracy is also improved.
> We feel quite excited about this result.
>
> If we have applications such as face recognition in mind, then the proper baseline is the Cosine Softmax, where the state-of-the-art face recognition algorithms such as CosFace (Wang et al. (2018)) and ArcFace (Deng et al. (2019)) are based on.
> Compared with Cosine Softmax, we observe improvements of our approach on all dimension settings (see Table 1 in manuscript).
> So it is promising to be applicable to such applications.
>
> ### Q2
>
> > If computation is not an issue, traditionally there are also methods in expanding the class representative vectors, e.g. multiple experts and fusion. It is not clear if the improvement is due to enlarged classification parameters or due to the new learning framework. Therefore, it would be better to show a comparison against multiple experts.
>
> Thanks for the question.
> We implement a ResNet50-V1D network with multiple independent classification fc layers and train them simultaneously.
> Their losses are averaged during training, and their logits are averaged during testing.
> The result is shown in Tab. 4-1 and we observe a performance drop for this setting.
> We analyze the learned weights for these fc layers and found that they basically converged to the same class representative vectors.
> This means that to improve the performance with multiple experts, a more sophisticated algorithm design is needed.
>
> Table 4-1. Compare Grassmannian ResNet50-V1D with Dim=8 on ImageNet with multi-experts.
>
> | Setting             | Parameters      | Top-1 | Top-5 |
> | ------------------- | --------------- | :---: | ----: |
> | Grassmannian (Ours) | Dim 8 subspaces | 79.12 | 94.41 |
> | Multi-Experts       | 8 fc experts    | 77.34 | 93.65 |
>
> ### Q3
>
> > The FN loss contributes a lot for the improvements. However, it should be a general trick that can also be applied on the traditional softmax baselines, which should also be reported for a fair comparison. Without the FN loss it appears that the improvement of the proposed method is still limited.
>
> Thanks for bringing out this issue.
> Below we add the requested experiment in Tab. 4-2.
> We find that the FN trick also works on the Cosine Softmax baseline.
> Note that the trick is not applicable to the vanilla softmax.
>
> Table 4-2. ResNet50-V1D on ImageNet using Cosine Softmax with FN regularization
>
> | Setting             | Top-1 | Top-5 |
> | ------------------- | :---: | ----: |
> | Cosine Softmax      | 78.30 | 94.07 |
> | Cosine Softmax + FN | 78.64 | 94.24 |

---

### Official Review · Reviewer_Ut4X · 2022-10-25

**Confidence:** 5
**Correctness:** 4
**Technical Novelty And Significance:** 2
**Empirical Novelty And Significance:** 3
**Recommendation:** 5

**Clarity, Quality, Novelty And Reproducibility:**

Clarity. This paper put their goal straightforward and clear.  The paper is very easy to follow.

Novelty, considering the reference [1], the novelty of the algorithm is limited. But I do appreciate that the paper finds good applications instead.

Reproducibility, it should be easy to reproduce the paper given the details.

**Strength And Weaknesses:**


Pros,

1. Replacing the de-facto combination of softmax and cross-entropy is interesting. They provide intensive experiments to validate their claim.

2. The experiments on the long-tail tasks provide a new sight to handle the data imbalance issue. It correlated to an extra memory bank or dictionary somehow.

Cons,
1. The authors miss an important reference [1]. The proposed method has been employed and used in subspace clustering instead of classification. The authors should clarify clearly your contribution and difference.


[1] Scalable Deep k-Subspace Clustering, Tong Zhang, Pan Ji, Mehrtash Harandi, Richard Hartley, Ian Reid, ACCV 2018.

Unclear parts:
1.  When there are more than two identical eigenvalues, there will be a sign flip issue in the corresponding eigenvectors. It may lead to the subspace update in a different way. Thus, I would like the authors to provide an analysis of such randomness due to subspace updating.

2. In table 2, as the Grassmannian with 16 dimensions is better than 8, why there is no FN experiment on 16 dimensions? I am also wondering how the sparsity is related to the FN. Comparing dimension 8, with FN and without FN the accuracy is quite similar but sparsity changes a lot.

3. The batch normalization will project the feature space to a unit sphere, which will be against the linear subspace assumption. Could the authors explain more on this direction and how you solve this issue?

4. Besides, I am also wondering how data augmentation affects the accuracy since the authors did not use augmentation in their implementation.


**Summary Of The Paper:**

This paper points out that using softmax does not take the intra-class and inter-class feature variation into account, and this paper aims to interpret the high dimensional feature output that lies in a union of linear subspaces. In classification, each feature representation falls into one of the subspaces (K classes), where each subspace is a Grassmannian manifold. To achieve this, this paper incorporates the Riemannian SGD into the ResNet50-D backbones to optimize the network and the k subspaces. The authors have validated that such an assumption is powerful and outperforms the softmax and cross-entropy combination in ImageNet-1K classification, feature transfer, and Long-tail classification.

**Summary Of The Review:**

Since the algorithm novelty of the paper is limited, I will need more details from the authors to justify whether they indeed bring new insights into this area.

---

> ### Author Response · Authors · 2022-11-16
> **Response to R3 (Part 2)**
>
> ### Q4
>
> > The batch normalization will project the feature space to a unit sphere, which will be against the linear subspace assumption. Could the authors explain more on this direction and how you solve this issue?
>
> Thanks for the question.
> BN does not cause trouble for the linear subspace assumption for the following reasons.
> (1) BN does not project the feature space to a unit sphere.
> BN normalizes the input element-wisely, so the whole feature, after BN, does not necessarily lie on a unit sphere.
> (2) The linear subspace assumption originates from the low-dimensional manifold assumption and linear subspace is one of the simplest manifolds we can use.
>
> ### Q5
>
> > Besides, I am also wondering how data augmentation affects the accuracy since the authors did not use augmentation in their implementation.
>
> Thanks for the question, augmentation indeed has an impact on performance.
> As described in the "Training Strategy" paragraph in Section 5.1, we use the standard augmentation for ImageNet, including random resized crop and random horizontal flip.
> We didn't test stronger augmentations in the manuscript because of limited computational resources at that time.
> Generally speaking, stronger augmentation mitigates the overfitting problem and benefits deep models.
> To demonstrate the effect of stronger augmentations, we run additional experiments in Tab. 3-2 using the RandAug (Cubuk et al. (2020)).
> We can see that stronger augmentation indeed further increases the accuracy.
> Together with longer training, the top-1 accuracy for ResNet50-V1D reaches 80.17%.
>
> Table 3-2. Grassmannian ResNet50-V1D on ImageNet with stronger augmentations
>
> | Setting        | Dim | Augmentation | Epochs | Top-1 | Top-5 |
> | -------------- | --- | ------------ | :----: | :---: | ----: |
> | Softmax        | -   | Standard     |  100   | 78.04 | 93.89 |
> | Softmax        | -   | RandAug      |  100   | 78.04 | 94.05 |
> | Cosine Softmax | -   | Standard     |  100   | 78.30 | 94.07 |
> | Cosine Softmax | -   | RandAug      |  100   | 78.95 | 94.55 |
> | Grassmannian   | 8   | Standard     |  100   | 79.12 | 94.41 |
> | Grassmannian   | 8   | RandAug      |  100   | 79.49 | 94.64 |
> | Grassmannian   | 8   | RandAug      |  300   | 80.03 | 94.77 |
> | Grassmannian   | 16  | Standard     |  100   | 79.21 | 94.29 |
> | Grassmannian   | 16  | RandAug      |  100   | 79.53 | 94.58 |
> | Grassmannian   | 16  | RandAug      |  300   | 80.17 | 94.93 |
>
> ### References
>
> - Cubuk, Ekin D., et al. "Randaugment: Practical automated data augmentation with a reduced search space." Proceedings of the IEEE/CVF conference on computer vision and pattern recognition workshops. 2020.

---

> ### Author Response · Authors · 2022-11-16
> **Response to R3 (Part 1)**
>
> We would like to thank the reviewer for the in-depth comments and suggestions.
> We answer the raised questions below.
>
> ### Q1
>
> > The authors miss an important reference [1]. The proposed method has been employed and used in subspace clustering instead of classification. The authors should clarify clearly your contribution and difference.
>
> Thanks for pointing out the reference and we will discuss it in the related work section.
> Note that there are major differences between the two works.
>
> 1. On formulation. The task is different. One is classification and the other is clustering.
>    The problem setting (supervised vs. unsupervised) and evaluation protocols are vastly different.
> 2. On optimization. The cluster subspace is updated at the end of every epoch, and the update is independent of the network parameters.
>    The class subspaces in our algorithm are updated simultaneously with network parameters in the same forward-backward pass at every iteration.
>    Besides, the subspace representing clusters in [1] requires good initialization from pre-trained networks, while ours can work with random initialization.
> 3. On application.
>    We conducted large-scale experiments on the ImageNet dataset, which contains 1.2M training images and consists of 1000 classes, while [1] is verified on small-scale datasets such as MNIST and Fashion-MNIST.
>
> ### Q2
>
> > When there are more than two identical eigenvalues, there will be a sign flip issue in the corresponding eigenvectors. It may lead to the subspace update in a different way. Thus, I would like the authors to provide an analysis of such randomness due to subspace updating.
>
> Thanks for the question.
> The subspace update does not depend on the sign choice of singular vectors in SVD.
> In Equ. (3), neither the term $\boldsymbol{V}\cos(t\boldsymbol{\Sigma})\boldsymbol{V}^T = \sum_i \cos(t\sigma_i) \boldsymbol{v}_i \boldsymbol{v}_i^T$ nor $\boldsymbol{U}\sin(t\boldsymbol{\Sigma})\boldsymbol{V}^T = \sum_i \sin(t\sigma_i) \boldsymbol{u}_i \boldsymbol{v}_i^T$ would be affected by the flip of sign since they come in pair and the signs of singular vectors will be canceled out during matrix multiplication.
>
> ### Q3
>
> > In table 2, as the Grassmannian with 16 dimensions is better than 8, why there is no FN experiment on 16 dimensions? I am also wondering how the sparsity is related to the FN. Comparing dimension 8, with FN and without FN the accuracy is quite similar but sparsity changes a lot.
>
> Thanks for the question.
> It was due to our computational resources at that time, we did most exploratory experiments on Dim=8 and Dim=1.
> We provide the complete results in Tab. 3-1.
> From the results, we can observe that FN works for the baseline Cosine Softmax as well.
> For Grassmannian + FN, there is an anomaly data point at Dim=16.
> It is possible that the FN trick does not work with very high subspace dimensions.
> We will add these results to the appendix.
>
> Table 3-1. Grassmannian ResNet50-V1D on ImageNet with FN regularization
>
> | Setting             | Dim | Top-1 | Top-5 |
> | ------------------- | --- | :---: | ----: |
> | Cosine Softmax + FN | -   | 78.64 | 94.24 |
> | Grassmannian + FN   | 1   | 78.65 | 94.24 |
> | Grassmannian + FN   | 2   | 78.87 | 94.43 |
> | Grassmannian + FN   | 4   | 79.10 | 94.58 |
> | Grassmannian + FN   | 8   | 79.37 | 94.53 |
> | Grassmannian + FN   | 16  | 79.09 | 94.37 |
>
> For the effect of FN on the sparsity of features, one possible explanation (based on observation) is that the FN loss encourages the norm of the feature vector (before re-normalization) to be larger than a threshold, one way to increase the norm is to have more non-zero activations, and the learning algorithms choose to do so.

---

### Official Review · Reviewer_iWx6 · 2022-10-26

**Confidence:** 3
**Correctness:** 3
**Technical Novelty And Significance:** 2
**Empirical Novelty And Significance:** 3
**Recommendation:** 6

**Clarity, Quality, Novelty And Reproducibility:**

-The paper is well written and easy to read. Related work is fairly covered.

- The idea of replacing fully connected layer with a geometric layer and the resulting impact on transfer learning and long tail classification is an interesting technical contribution.

- The authors have promised to released the code for reproducibility and also provided enough technical details in the submission.

**Strength And Weaknesses:**

Strength:

-Presentation: The submission is easy to read and follow. It is well written with intuitions provided where necessary. The problem is well motivated and contextualized in the broader scientific context.

-Technically solid and grounded work.


-Interesting empirical results wrt baselines considered in the submission and thus a promising direction.


Weakness:

- Technical Novelty, from Deep (Riemannian) Manifold Learning perspective, is somewhat marginal.

- Experimental Validation is heavily centered around Grassmanian baselines. While this submission cites several works that explicitly encourage/promote intra-class variablity, comparison with such baselines is completely missing.

- While the problem of promoting intra class variability is of great interest in deep learning, the proposed method does not explicitly model it as such. I concede it is not a strong weakness and based on the empirical findings in this work, future work can address this explicitly.

**Summary Of The Paper:**

The overall goal of the submission is a learning formulation that simultaneously models inter-class discrimination while promoting intra class variation in classification. To this end, it considers a linear subspace approach that is scaleless and thus in theory more suitable for long tail classification than the vector counterpart. Since set of subspaces form a Grassmann manifold, the submission replaces the fully connected layer in deep networks with a geometric one and optimizes it with Riemannian SGD. The method is validated on various benchmarks where it is shown to improve the vanilla baseline on transfer learning as well as long tail classification tasks.

**Summary Of The Review:**

The approach is based on a principled framework even though the technical novelty is not very strong from a broader manifold learning perspective. The resulting gain wrt Grassmanian and vanilla baselines are interesting even though the method does not explicitly model the intra class variability in the formulation unlike some existing work in this direction which are not compared with.

---

> ### Author Response · Authors · 2022-11-16
> **Response to R2**
>
> We would like to thank the reviewer for the valuable comments and suggestions on the manuscript.
> We address the concerns below.
>
> ### Q1
>
> > Technical Novelty, from Deep (Riemannian) Manifold Learning perspective, is somewhat marginal.
>
> We thank the reviewer for raising this issue.
> We do not intend to modify the optimization procedures of manifold learning, rather, we mainly apply the manifold learning in deep networks to learn the subspace representation of classes.
> To the best of our knowledge, this is the first time that manifold learning achieved competitive results on large-scale datasets as ImageNet.
>
> ### Q2
>
> > Experimental Validation is heavily centered around Grassmannian baselines. While this submission cites several works that explicitly encourage/promote intra-class variability, comparison with such baselines is completely missing.
>
> Thanks for raising this issue.
> The mentioned works do encourage feature diversity, however, they are not proper baselines as we will detail below.
> The SoftTriple loss (Qian et al. (2019)) is designed for deep metric learning and does excel on fine-grained datasets under the NMI and Recall@K metrics.
> However, their networks need to be initialized with a model pre-trained on the full ImageNet dataset.
> We adapted their code and trained on ImageNet from scratch directly, the result (Tab. 2-1) is not competitive.
> The global orthogonal regularization (Zhang et al. (2017)) increases feature diversity by encouraging **inter-class** diversity (regularize only on non-matching pairs) and it works only for triplet loss.
> So it is not a proper baseline of ours.
>
> Table 2-1. Compare Grassmannian ResNet50-V1D with Dim=8 on ImageNet with SoftTriple.
>
> | Setting             | Parameters           | Top-1 | Top-5 |
> | ------------------- | -------------------- | :---: | ----: |
> | Grassmannian (Ours) | Dim 8 subspaces      | 79.12 | 94.41 |
> | SoftTriple          | 8 centers each class | 75.55 | 92.62 |
>
> ### Q3
>
> > While the problem of promoting intra class variability is of great interest in deep learning, the proposed method does not explicitly model it as such. I concede it is not a strong weakness and based on the empirical findings in this work, future work can address this explicitly.
>
> We appreciate the reviewer’s suggestion.
> As noted in Section 6 on limitations and future directions, we plan to combine it with self-supervised learning to explicitly encourage intra-class variabilities.
> It will be done in a separate work.
>
> ### References
>
> - Qian, Qi, et al. "Softtriple loss: Deep metric learning without triplet sampling." Proceedings of the IEEE/CVF International Conference on Computer Vision. 2019.
> - Zhang, Xu, et al. "Learning spread-out local feature descriptors." Proceedings of the IEEE international conference on computer vision. 2017.

---

### Official Review · Reviewer_K7X9 · 2022-10-26

**Confidence:** 4
**Clarity, Quality, Novelty And Reproducibility:** As I mentioned above, I think the pap…
**Correctness:** 3
**Technical Novelty And Significance:** 3
**Empirical Novelty And Significance:** 3
**Recommendation:** 6

**Strength And Weaknesses:**

The strengths:
The paper is clearly written and well-motivated. The method is sufficiently described for a reader's implementation. The experiments are diverse: Beyond testing accuracy on ImageNet, the paper also studies feature transfer to new datasets as well as learning from small number of examples.

The weaknesses: Some of the improvements demonstrated in experiments are fairly small (e.g. Table 1). In fact, given the variability in numbers one might get in all the experiments through small tinkering with known tricks, it is difficult to know if the demonstrated advantages would hold under further optimization for any given task (and if they are statistically significant across initializations in learning), although experiments and illustrations as a whole do paint a convincing picture. Some of the choices of the implementation are not fully explained.

Overall, my first impression is positive.

Questions:
1. Given the need for step 5 in the algorithm (orthogonalization) for numerical stability, why bother with the geodesic at all and simply do gradient descent on S followed by step 5?  Are there some illustrations of why it fails?
2. Just looking at Eq. 6, one might ask if the orthogonal subspace basis and Grassman manifold view is really necessary, or if the benefit simply comes from the quadratic form of the logit computation (instead of linear). I.e., going beyond the previous question: Can the optimization be simply done on unconstrained S_i? Or for that matter can the logit be l_i=x' W x, with unconstrained W (gradient descent optimization of W tends to regularize it to be low rank anyhow).
3. Given the quadratic nature of the computation, is there a relationship to tensor product representations (Smolenski et al), where such computations are done in all layers of a network? (and do you plan to move your subspace projection into earlier layers, too?)
4. Norm regularization (12), as well as in equation 4 (to renormalize x to constant \gamma norm) may play big roles in learning reducing the real effect of the subspace modeling (and also do you do both of these things or just (12)?)
5. In the first ImageNet experiment, how would you account for the change in the modeling power by simply having more parameters in the last layer?
6. In the transfer experiments, I am assuming that the issue above no longer exists, because you treat the features from the previous layer the same way (i.e. not through fine-tuned subspace projections, but using a linear classifier).  Is that right?
7. If the above is right, then Table 2 may be slightly confusing, as results for ImageNet seem to be copied from Table 1, where logits a computed using norm of the subspace projections, but for the rest of the datasets, they are computed using linear projections.
8. Finally, the premise of the experiments is that the joint training of the backbone and the (subspace-based) classifier results in features that are better in the ways described in the paper.  If you initialize the network trained with regular softmax or cosine softmax classifier layer, and then switch to the subspace-based layer, what happens? Can keeping the features fixed and finding good subspaces increase accuracy? Does further training of the network change the features and how? (or is this not a meaningful experiment because of the lack of the bias term in your model?)

**Summary Of The Paper:**

The last layer of a neural network trained to classify instances usually linearly projects the feature vector from the previous layer to compute log odds of a class. This paper proposes replacement of that linear projection with a norm of the projection of the feature vector into a subspace. The paper shows how to optimize all weights of such a network and studies the advantages of the representation. The authors argue that the learned features transfer better in downstream tasks.

**Summary Of The Review:**

While I do have questions, I also like the discussion of the ideas in the paper. I don't regret reading it :), so I imagine others would not, either. So, unless something essentially equivalent has already been done (and I am not aware of it), then publication is justified. I am looking forward to the authors' response so I can understand the ideas and the details even better.

---

> ### Author Response · Authors · 2022-11-16
> **Response to R1 (Part 3)**
>
> ### Q6
>
> > In the transfer experiments, I am assuming that the issue above no longer exists, because you treat the features from the previous layer the same way (i.e. not through fine-tuned subspace projections, but using a linear classifier). Is that right?
>
> Yes, for the transfer learning experiment, we use the features before the last classification layer.
> This follows the testing protocol of Kornblith et al. (2021).
> The features are all fixed and a new linear SVM classifier is trained on these fixed features for each transfer learning testing dataset.
>
> ### Q7
>
> > If the above is right, then Table 2 may be slightly confusing, as results for ImageNet seem to be copied from Table 1, where logits a computed using norm of the subspace projections, but for the rest of the datasets, they are computed using linear projections.
>
> The ImageNet results in Table~2 are not part of the transfer learning results, but for reference only.
> Both the top-1 accuracy and the $R^2$ scores are useful side information that helps the understanding of feature transfer performance.
>
> ### Q8
>
> > Finally, the premise of the experiments is that the joint training of the backbone and the (subspace-based) classifier results in features that are better in the ways described in the paper. If you initialize the network trained with regular softmax or cosine softmax classifier layer, and then switch to the subspace-based layer, what happens? Can keeping the features fixed and finding good subspaces increase accuracy? Does further training of the network change the features and how? (or is this not a meaningful experiment because of the lack of the bias term in your model?)
>
> Thanks for the question.
> Joint training is essential for learning better features.
> To support this claim, we add the suggested two fine-tuning experiments in Tab. 1-5.
> The first setting is to fine-tune the last geometric fc layer from the weights trained from regular softmax (the top-1 is 78.04%), with all other parameters frozen.
> And the second setting fine-tunes all parameters.
> From results in Tab. 1-5, we find that if the feature is fixed, changing the regular fc to the geometric version does not increase performance noticeably (from 78.04% to 78.14%).
> But when all parameters are free to learn, then the pre-trained model is a better initialization than the random initialization (top-1 from 79.12% to 79.44%).
>
> Table 1-5. Compare Grassmannian ResNet50-V1D with Dim=8 on ImageNet with fine-tuning.
>
> | Setting                              | Initialization                       | Top-1 | Top-5 |
> | ------------------------------------ | ------------------------------------ | :---: | ----: |
> | Vanilla Softmax                      | Random                               | 78.04 | 93.89 |
> | Grassmann, Train From Scratch (Ours) | Random                               | 79.12 | 94.41 |
> | Grassmann, Fine-tune Only Last Layer | Weights trained from vanilla softmax | 78.14 | 93.97 |
> | Grassmann, Fine-tune All Parameters  | Weights trained from vanilla softmax | 79.44 | 94.58 |
>
> ### References
>
> - Absil, P-A., Robert Mahony, and Rodolphe Sepulchre. "Optimization algorithms on matrix manifolds." Optimization Algorithms on Matrix Manifolds. Princeton University Press, 2009.
> - Qian, Qi, et al. "Softtriple loss: Deep metric learning without triplet sampling." Proceedings of the IEEE/CVF International Conference on Computer Vision. 2019.
> - Kornblith, Simon, et al. "Why do better loss functions lead to less transferable features?." Advances in Neural Information Processing Systems 34 (2021): 28648-28662.

---

> ### Author Response · Authors · 2022-11-16
> **Response to R1 (Part 2)**
>
> ### Q3
>
> > Given the quadratic nature of the computation, is there a relationship to tensor product representations (Smolenski et al), where such computations are done in all layers of a network? (and do you plan to move your subspace projection into earlier layers, too?)
>
> Thanks for the question, our responses are divided into two parts.
>
> **Relationship to tensor product representations**
>
> Although the matrix representation $\boldsymbol{S}\_i$ for class $i$ is an order-2 tensor mathematically, after a preliminary reading of the tensor product representation (Smolenski et al), we did not find similarity between the two.
> The matrix form here is a way to represent the geometric object, namely the linear subspace, and the matrix is even not unique for the same subspace.
>
> **Move your subspace projection into earlier layers**
>
> We do not plan to move it into earlier layers.
> The motivation of this work is to model classes as subspaces.
> There are yet no such concrete concepts (e.g. classes) we want to model in earlier layers.
> Nevertheless, we want to mention that applying geometric constraints on general layers is the topic of orthogonal constraints which we have briefly reviewed in the related work section.
>
> ### Q4
>
> > Norm regularization (12), as well as in equation 4 (to re-normalize x to constant $\gamma$ norm) may play big roles in learning reducing the real effect of the subspace modeling (and also do you do both of these things or just (12)?)
>
> Thanks for raising the issue.
> For all experiments, we re-normalizes the feature vector (including experiments with "+FN"), and only experiments marked with "+FN" has norm regularization (12).
> We will make this statement more visible in the main text.
> Norm regularization is applied to features just before the re-normalization, so they can work together.
> Their roles are different.
> Feature re-normalization is to ignore the length of the feature, while norm regularization (12) is to avoid the features before normalization being too small.
> The feature re-normalization is essential for effective learning of subspaces while the "FN" is a good-to-have trick.
> For reference, the Cosine softmax also requires feature re-normalization for effective learning.
> We add experiments that compare with/without feature re-normalization in Tab. 1-3.
>
> Table 1-3. Grassmannian ResNet50-V1D with Dim=1,8 on ImageNet with/without feature re-normalization
>
> | Setting                  | Dim | Top-1 | Top-5 |
> | ------------------------ | --- | :---: | ----: |
> | With Re-normalization    | 1   | 78.48 | 94.24 |
> | Without Re-normalization | 1   | 77.91 | 93.78 |
> | With Re-normalization    | 8   | 79.12 | 94.41 |
> | Without Re-normalization | 8   | 78.12 | 93.90 |
>
> ### Q5
>
> > In the first ImageNet experiment, how would you account for the change in the modeling power by simply having more parameters in the last layer?
>
> Thanks for asking.
> More parameters in the last layer enhance the modeling power.
> For example, in the experiment section, we demonstrated that class relationships could be richer than a single angle.
> For comparison with other methods that use the same amount of parameters, we add two experiments in Tab. 1-4.
> One uses multiple fc as multi-experts.
> When training, these independent fcs are trained side by side, and their losses are averaged.
> When testing, the logits are first averaged, and then output the prediction after softmax.
> The other one is the SoftTriple loss (Qian et al. (2019)), where each class is modeled by multiple centers.
> For each class, the logit is the weighted combination of logits computed from individual centers.
> All training protocols such as augmentation, learning rate schedule, etc. are the same as ours.
> For results in Tab. 1-4, we found that the Grassmannian class representation is the most effective one.
>
> Table 1-4. Compare Grassmannian ResNet50-V1D with Dim=8 on ImageNet with multi-experts and SoftTriple.
>
> | Setting             | Parameters           | Top-1 | Top-5 |
> | ------------------- | -------------------- | :---: | ----: |
> | Grassmannian (Ours) | Dim 8 subspaces      | 79.12 | 94.41 |
> | Multi-Experts       | 8 fc experts         | 77.34 | 93.65 |
> | SoftTriple          | 8 centers each class | 75.55 | 92.62 |

---

> ### Author Response · Authors · 2022-11-16
> **Response to R1 (Part 1)**
>
> We would like to thank the reviewer for the thoughtful and thorough review and the comments and suggestions are invaluable for the improvement of the manuscript.
> Below we address the concerns in detail.
>
> ### Q1
>
> > Given the need for step 5 in the algorithm (orthogonalization) for numerical stability, why bother with the geodesic at all and simply do gradient descent on S followed by step 5? Are there some illustrations of why it fails?
>
> Thanks for the good question, and your suggestion is perfectly valid.
> We explain it in two parts.
>
> **Numerical stability**
>
> The numerical stability issue is caused by the accumulation of tiny computational errors of Equ. (3).
> After many iterations, the resultant matrix $\boldsymbol{S}$ might not be perfectly orthogonal.
> For example, after 100, 1000, and 5000 iterations (subspace dim=8), we observed that the error $\max_i\lVert\boldsymbol{S}\_i^T\boldsymbol{S}\_i-\boldsymbol{I}\rVert{}_\infty$ is 1.9e-5, 9.6e-5 and 3.7e-4, respectively.
> After 50 epochs, the error accumulates to 0.0075, but we did not find any noticeable influence on the performance yet.
> For this reason, we marked this step as "optional" in the manuscript.
>
> **Simply do gradient descent followed by step 5**
>
> Replacing Step 4 with the Euclidean gradient update and using the re-orthogonalization by QR decomposition in Step 5 will also work.
> The QR operation is a "retraction" in Geometric optimization, which is an approximate way of moving along a geodesic (details see Absil et al. (2009), Equ. (4.11)).
> So after the modification, it becomes another version of Riemannian SGD (we call it Alg. 1 variant in Tab. 1-1).
> To verify the validity of the QR retraction version of Riemannian SGD, we implemented it and use it to train the Grassmannian ResNet50-V1D with Dim=8 on ImageNet.
> The result is shown in Table 1-1 below, and we can see that the proposed class representation works on both implementations of Riemannian SGD.
>
> Table 1-1. Grassmannian ResNet50-V1D with Dim=8 on ImageNet with different retraction strategies
>
> | Riemannian SGD | Retraction            | Top-1 | Top-5 |
> | -------------- | --------------------- | :---: | ----: |
> | Alg. 1         | moving along geodesic | 79.12 | 94.41 |
> | Alg. 1 Variant | retraction by QR      | 79.13 | 94.45 |
>
> ### Q2
>
> > Just looking at Eq. 6, one might ask if the orthogonal subspace basis and Grassmann manifold view is really necessary, or if the benefit simply comes from the quadratic form of the logit computation (instead of linear). I.e., going beyond the previous question: Can the optimization be simply done on unconstrained S_i? Or for that matter can the logit be l_i=x' W x, with unconstrained W (gradient descent optimization of W tends to regularize it to be low rank anyhow)
>
> Thanks for the question.
> The Grassmannian view is essential, as we will demonstrate in the experiments in Tab. 1-2.
> We trained ResNet50-V1D with Dim=8 on ImageNet.
> Firstly we remove the constraint on $\boldsymbol{S}\_i$ by optimizing directly using the vanilla SGD (the "Unconstrained" setting in Tab. 1-2).
> Then we further change the formula of logit to the quadratic form (the "Unconstrained Square" setting in Tab. 1-2).
> Note that the expression $\boldsymbol{x}' \boldsymbol{W} \boldsymbol{x}$ is invalid due to dimension mismatch, so we used the squared norm $\lVert\boldsymbol{S}\_i^T \boldsymbol{x}\rVert^2=\boldsymbol{x}^T \boldsymbol{S}\_i \boldsymbol{S}\_i^T \boldsymbol{x} $ instead.
> We observed a performance drop for both changed settings.
>
> Table 1-2. Grassmannian ResNet50-V1D with Dim=8 on ImageNet with unconstrained $\boldsymbol{S}\_i$
>
> | Setting              | Logit                                         | Optimizer | Top-1 | Top-5 |
> | -------------------- | --------------------------------------------- | --------- | :---: | ----: |
> | Alg. 1               | Equ. (6): $\lVert\boldsymbol{S}\_i^T x\rVert$ | RSGD      | 79.12 | 94.41 |
> | Unconstrained        | $\lVert\boldsymbol{S}\_i^T x\rVert$           | SGD       | 78.55 | 94.18 |
> | Unconstrained Square | $\lVert\boldsymbol{S}\_i^T x\rVert^2$         | SGD       | 77.09 | 93.50 |

---

### Author Response · Authors · 2022-11-16
**Thanks for the valuable comments and suggestions!**

Firstly, we would like to thank all the reviewers for their thorough and thoughtful comments on the manuscript.
And we also sincerely apologize for the relatively late response.
We just finished running 23 additional experiments that help address the reviewers' concerns.
The results are included in the following tables.

- Tab. 1-1. Comparison with alternative retraction
- Tab. 1-2. Comparison with unconstrained parameters
- Tab. 1-3. Comparison with re-normalization off
- Tab. 1-4. Comparison with multi-experts and SoftTriple
- Tab. 1-5. Comparison with fine-tuning
- Tab. 2-1. Comparison with SoftTriple (subset of Tab. 1-4)
- Tab. 3-1. Comparison with FN regularization on more dims
- Tab. 3-2. Comparison with stronger augmentations
- Tab. 4-1. Comparison with multi-experts (subset of Tab. 1-4)
- Tab. 4-2. Comparison with FN on baseline (subset of Tab. 3-1)
- Tab. 4-3. Comparison with adding margins
- Tab. 5-1. Comparison with non-learnable subspaces (subset of Tab. 1-5)
- Tab. 5-2. Comparison with iteration time
- Tab. 5-3. Comparison with SVD time
- Tab. 5-4. Comparison with ResNet101-V1D and RexNeXt50.

Note that each individual experiment requires a full training on the ImageNet dataset, which typically needs 1-2 days on an 8xGPU server.
So it took us a lot of time to process.
A highlight of these results is that with stronger augmentations and longer training, the top-1 accuracy of our formulation can achieve 80.17% (details in Tab. 3-2).
We will add these supplementary results to the appendix, and we believe the valuable suggestions from reviewers will make the manuscript more solid.
Again, thanks for your patience and we appreciate your further feedback.

---

### Author Response · Authors · 2022-11-18
**Revision Summary**

We would like to thank the reviewers again for their valuable suggestions and detailed comments.
We made minor edits to the main text and major revisions to the appendix to clarify ambiguities and address reviewers' concerns.
The notable changes are:

1. We added discussions of Zhang et al. (2018), Simon et al. (2020), Roy et al. (2019), and Harandi & Fernando (2016) in the related work section.
2. We added "To simplify notation, we assume feature $\boldsymbol{x}$ has been properly re-normalized throughout this paper unless otherwise specified." in Section 4 to clearly state that features are re-normalized.
3. We added a detailed caption for Figure 1.
4. We added a new section "Technical Details" in the appendix to discuss the (a) Alternative Implementation of Riemannian SGD, (b) Necessity of Grassmannian Formulation and Geometric Optimization, and (c) Numerical Stability of Algorithm 1
5. We enriched the "Hyper-Parameters and Design Decisions" section in the appendix to discuss (a) Importance of Joint Training, (b) More Results of FN and (c) Stronger Augmentation Improves Accuracy
6. We added a new section "More Baselines" in the appendix to discuss the results of Multi-FC, SoftTriple, ResNet101, and ResNeXt50.
7. We revised the "Training Speed and SVD speed" section in the appendix to give more detailed measurements and benchmarks on running time and discussed the speed of SVD with more detail.

---

### Decision · Program_Chairs · 2023-01-20

**Decision:**

Reject

**Justification For Why Not Higher Score:**

After rebuttal and AC-rev. discussions, several issues as described above lingered (issues with the limited novelty, limited evaluations, and marginal improvements).

**Justification For Why Not Lower Score:**

N/A

**Metareview: Summary, Strengths And Weaknesses:**

This paper was evaluated by five reviewers and received 2x5 and 3x6 borderline scores. While the reviewers liked the paper they had also a number of important concerns that remained including lack of strong comparisons with existing bilinear methods, matrix square root methods, log-Euclidean and other non-Euclidean classifiers [A,B,C] (there are many more works than A B C). Reviewers also noted improvements were marginal and evaluations could have been conducted on a larger number of datasets often used with the above methods. Also, vector-to-subspaces has been used before as a classifier in several works, for example in FSL [D,E,F]. Overall, this is a well established concept and as such the paper requires non-incremental contributions, evaluations and findings to be accepted. On this occasion, the paper is below the acceptance bar and requires a new round of reviews after fixing the above issues.

A. Improved Bilinear Pooling with CNNs, Lin et al.
B. Fast Differentiable Matrix Square Root, Song et al.
C. Why Approximate Matrix Square Root Outperforms Accurate SVD in Global Covariance Pooling? Song t al.
D. Regression Networks for Meta-Learning Few-Shot Classification, Devos  et al.
E. Adaptive Subspaces for Few-Shot Learning, Simon et al.
F. TapNet: Neural Network Augmented with Task-Adaptive Projection for Few-Shot Learning, Yoon et al.

**Summary Of Ac-Reviewer Meeting:**

As noted above, lacks of thorough comparisons with existing bilinear methods, matrix square root methods, log-Euclidean and other non-Euclidean classifiers [A,B,C] (there are many more works than A B C) was noted. Marginal improvements were noted, and lack of extensive evaluations on a larger number of datasets often used with the above methods. It was noted that vector-to-subspaces distance has been used before as a classifier in several works (see metareview for the list). Overall, subspaces is a well established concept and as such the paper requires non-incremental contributions, evaluations and findings to be of sufficient novelty. The meeting was a mix of in-person attendees and reviewers who messaged in their thoughts directly.